# Observations of preferential summer melt of Arctic sea-ice ridge keels from repeated multibeam sonar surveys

Evgenii Salganik[1,2], Benjamin A. Lange[1,3], Christian Katlein[4], Ilkka Matero[4,5], Philipp Anhaus[4], Morven Muilwijk[1], Knut V. Høyland[2], Mats A. Granskog[1]

[1]Norwegian Polar Institute, Tromsø, 9296, Norway
[2]Norwegian University of Science and Technology, Trondheim, 7491, Norway
[3]Norwegian Geotechnical Institute, Oslo, 0484, Norway
[4]Alfred-Wegener-Institut Helmholtz-Zentrum für Polar- und Meeresforschung, Bremerhaven, 27570, Germany
[5]Svalbard Integrated Arctic Earth Observing System Knowledge Centre, Longyearbyen, 9171, Svalbard

*Correspondence to*: Evgenii Salganik (evgenii.salganik@proton.me)

**Abstract.** Sea-ice ridges constitute a large fraction of the total Arctic sea-ice area (up to 40–50 %); nevertheless, they are the least studied part of the ice pack. Here we investigate sea-ice melt rates using rare, repeated underwater multibeam sonar surveys that cover a period of one month during the advanced stage of sea-ice melt. Bottom melt increases with ice draft for first- and second-year level ice and a first-year ice ridge, with an average of 0.46 m, 0.55 m, and 0.95 m of total snow and ice melt in the observation period, respectively. On average, the studied ridge had a 4.6 m keel bottom draft, was 42 m wide, and had 4 % macroporosity. While bottom melt rates of ridge keel were 3.8 times higher than first-year level ice, surface melt rates were almost identical but responsible for 40 % of ridge draft decrease. Average cross-sectional keel melt ranged from 0.2 m to 2.6 m, with a maximum point ice loss of 6 m, showcasing its large spatial variability. We attribute 57 % of the ridge total (surface and bottom) melt variability to keel draft (36 %), slope (32 %), and width (27 %), with higher melt for ridges with a larger draft, a steeper slope, and a smaller width. The melt rate of the ridge keel flanks was proportional to the draft, with increased keel melt within 10 m of its bottom corners and the melt rates between these corners comparable to level ice melt.

## 1 Introduction

According to the definition by the World Meteorological Organization, an ice ridge is a line or wall of broken ice that is forced up by pressure (WMO, 2014). Ridges consist of a sail above and a keel below the water level. The keel initially consists of randomly packed ice blocks separated by water-filled voids, described by the ridge macroporosity (fraction of water-filled voids in the keel). The initial macroporosity of first-year ice ridges is in the range of 20 % to 45 % (Bowen and Topham, 1996), with an average porosity of 30 % (Timco and Burden, 1997). The upper part of ridge keels usually refreezes, forming a consolidated layer defined by zero macroporosity. Some ridges become fully consolidated (with near-zero keel macroporosity) during the melt season (Marchenko, 2022). The measurements collected in the Artic Ocean during the Multidisciplinary drifting Observatory for the Study of the Arctic Climate (MOSAiC) expedition (Nicolaus et al., 2022) showed that complete consolidation of ridges may occur during the spring season before the melt onset through the transfer of snow into ridge keels

via open leads (Salganik et al., 2023a) or ice deformation, which was supported by 6–11 % estimates of snow mass fraction within several ridges (Lange et al., 2023). Ice ridges are key features in climate studies since they constitute around 30 % of the total Arctic sea-ice volume based on ice-ocean coupled modelling (Rothrock, 2005). While Mårtensson et al. (2012) used a multicategory sea ice model to estimate the Arctic sea-ice ridge volume of 45–60 % and ridge area of 25–45 %. Melling & Riedel (1996) observed an increase in ridge areal fraction from 15 % in autumn to 40–50 % in spring based on subsea sonar ice draft measurements in the Beaufort Sea in 1991–1992. However, the proportion of ridges varies depending on the region and how they are defined. Fram Strait serves as the main outlet of the Arctic sea ice export (Krumpen et al., 2016), and for that region, Hansen et al. (2014) estimated the fraction of deformed ice of 37±8 %, using an evolving threshold relative to the modal thickness derived from draft measurements by moored upward-looking sonars during 1990–2011. In those observations, the ridge fraction increased in 1990–2008 and decreased thereafter, which was confirmed by Sumata et al. (2023) using extended data from the same upward-looking sonars in Fram Strait for 1990–2020. Furthermore, ridges have also been identified as potential biological hotspots (Gradinger et al., 2010; Fernández-Méndez et al., 2018) and as influencing the light conditions beneath the ice (Katlein et al., 2021).

Sea-ice ridges can be formed from new, young, first-year, second-year, or multiyear level ice, or from a combination of ice types. Typically, ridges are made from relatively thin ice (Tucker et al., 1984), which breaks as the weakest points during deformation events. Ridges themselves can also be first-year, second-year, or multiyear, depending on how many seasons they have survived. The maximum keel draft is limited by the ice strength and is correlated with the adjacent level ice draft (Amundrud et al., 2004). Once the keel has reached its maximum possible draft, it thereafter only grows in width (Hopkins, 1998).

Previous research has suggested that ridges impact the melt rates of the ice. For instance, Rigby and Hanson (1976) showed enhanced bottom melt of a ridge keel in comparison to thinner ice, although mechanical erosion could not be ruled out for this large ridge with a maximum total thickness of 10–12 m. During the SHEBA expedition in the Beaufort and Chukchi Seas, Perovich et al. (2003) used data from single-point measurements from hot-wire thickness gauges and measured 60 % higher bottom melt for second-year and multiyear ice ridges (42 gauges) than for multiyear level ice (89 gauges) during the entire melt season from early June to early October 1998. While, Skyllingstad et al. (2003) measured enhanced vertical mixing and a five-fold increase in ocean heat flux (OHF) for a 10-m-deep ridge during the winter season at SHEBA expedition using high-frequency measurements of seawater temperature, salinity, and velocity. The same effect is also likely in summer, but this does not take into account the shallow meltwater stratification that develops in summer and affects ice melt rates (Salganik et al., 2023b). Amundrud et al. (2006) estimated that ridge keels melt 4–5 times faster than level ice based on the observations from ice-profiling sonars mounted on subsea moorings in the Beaufort Sea (however, their data does not repeatedly measure the same ice due to sideway ice drift). Furthermore, Shestov et al. (2018) observed ridge melt in summer during the N-ICE2015 expedition (Granskog et al., 2018) in the pack ice north of Svalbard using single-point measurements from a temperature buoy. Here, the average OHF under level ice was 63 W m$^{-2}$ (Peterson et al., 2017), while the ridge keel melted by 1.5 m over two weeks, which translates into an equivalent OHF of 300 W m$^{-2}$ (with macroporosity of 27 % taken into account), 4.8 times

higher than for level ice (Shestov et al., 2018). Based on the thermodynamic model developed by Amundrud et al. (2006), several parameters, such as keel width and shape, may impact keel melt, with ridge macroporosity and block thickness being key factors. In summary, the observed ratio of ridge and level ice accumulated melt in previous studies ranged from 60 % to 400 % even for similar geographical locations, suggesting the need for a more detailed investigation of the spatial and temporal variability of melting of different ice types.

The first direct measurements of under-ice topography were linear profiles from narrow-beam upward-looking sonar (Lyon, 1961). Wadhams et al. (2006) and Wadhams & Doble (2008) were the first to use an autonomous underwater vehicle instrumented with a multibeam sonar to study the three-dimensional bottom topography of Arctic sea ice. Using multibeam mapping by a submarine, Wadhams & Toberg (2012) found a mean slope of first-year and multi-year ridge keels of 28° and 25°, respectively, assuming a triangular shape. Ekeberg et al. (2015) analysed the shape of ridge keels using data from upward-looking sonar in Fram Strait and suggested that ridge keels typically have a trapezoidal shape, with the keel bottom width accounting for an average of 17 % of the keel total width.

Although ridges play an important role in the evolution of the Arctic ice pack, ridges are understudied compared to the level ice that is usually sampled. The aforementioned studies are also typically limited to a one-time snapshot and a few point measurements. In this study, we use novel repeated multibeam ice draft measurements that follow the temporal and spatial evolution of a first-year sea-ice ridge and adjacent level ice during summer melt collected in the Arctic Ocean during the MOSAiC expedition in 2020. Over a period of a month, we observed ice draft changes and melt rates for first- and second-year level ice and a first-year ice ridge. Additionally, we identified key characteristics of the ice bottom topography that affected the melt rates. In the first two sections of this study, we provide estimates of the total melt for level ice and ridge; in the third section, we analyse the effect of ridge cross-sectional characteristics on its melt; in the fourth section, we provide estimates of surface and bottom melt for level ice and ridge; in the fifth and sixth sections, we discuss how meltwater drainage and sea-ice density temporal evolution affect the sea ice draft; and in the seventh and eighth sections, we compare our estimates of the ridge enhanced melt with previous observations and discuss the limitations of this study. The use of underwater multibeam sonar allows to collect over $10^5$ measurements of sea ice draft and over $10^4$ ridge draft measurements every week. In comparison to point measurements from temperature buoys (Shestov et al., 2018), ice coring, and thickness gauges (Perovich et al., 2003), ROV sonar measurements increase the number of draft data points by three orders of magnitude, revealing the small-scale spatial variability of sea ice melt in unprecedented detail. In comparison to moored subsea sonar ice draft measurements (Amundrud et al., 2006), repeated ROV surveys allowed us to study the same sea ice repeatedly for a longer period with high spatial resolution and reduced uncertainty in measured melt rates.

## 2 Materials and Methods

### 2.1 Expedition

The MOSAiC expedition took place in 2019–2020 to better understand the coupled Arctic ice, ocean, and atmosphere system and the sea-ice mass and energy budget over a full season and included a range of snow and sea ice measurements (Nicolaus et al., 2022). The MOSAiC Central Observatory, an approximately 3 km by 4 km large ice floe, drifted for 10 months across the central Arctic starting from 4 October 2019, following the Transpolar Drift, until it reached the ice edge in Fram Strait and broke apart on 31 July 2020 (Fig. 1b). The OHF estimate from ice mass balance buoys (IMBs) increased from 11 W m$^{-2}$ to 40 W m$^{-2}$ during July 2020 (Salganik et al., 2023b), with an average value of 24 W m$^{-2}$, comparable to the summer OHF estimates of 20–30 W m$^{-2}$ for Beaufort Gyre and Transpolar Drift in 1979–2002 (Krishfield, 2005). The OHF increase took place in mid-July, the period with the highest annual solar insolation, and with a reduction in sea ice concentration from 100 % to 85 % within a 3 km radius of the Central Observatory (Krumpen et al., 2021). This combined led to a higher solar heat input and hence warming of the mixed layer (Fig. 1d). Despite the floe also drifting further south towards Fram Strait and getting closer to shallower and warmer Atlantic Water at the same time, the mixed layer and upper ocean conditions still retained their Arctic characteristics. This is evident through the observation of remarkably low heat fluxes over the halocline (–0.01±0.30 W m$^{-2}$) and thermocline (+2.1±1.2 W m$^{-2}$) in Fram Strait during melt season (Schulz et al., 2023). This shows that the conditions are representative of the seasonally driven OHF typical for the central Arctic and are not driven by excessive OHF in the marginal ice zone or from Atlantic water, as observed by, e.g., Shestov et al. (2018).

Due to logistical reasons, the ridge investigations during MOSAiC were performed at several sites. During January–February and June–July 2020, Alli's Ridge was studied using ice coring and IMBs, while in June–July 2020, Jaridge was studied using underwater sonar surveys, ice coring, and IMB. The measurements of Alli's Ridge draft included only 4 ice coring sites visited before and during melt season (Salganik et al., 2023a) and will be used only as a reference for the Jaridge investigations presented here.

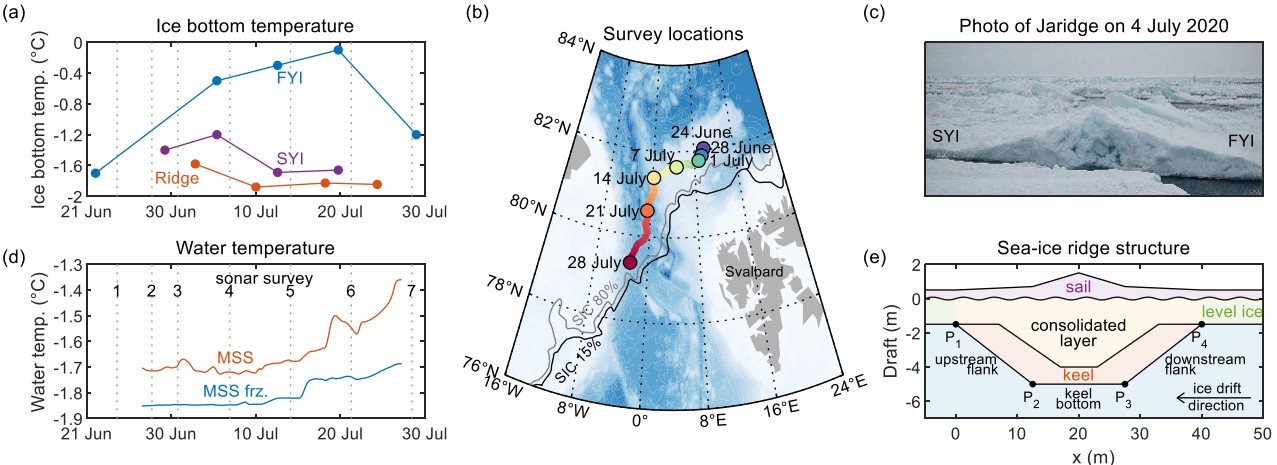

**Figure 1: (a)** Ice bottom temperature for first-year ice (FYI), second-year ice (SYI) and ridge from coring; **(b)** overview map of the study area with drift of the MOSAiC ice floe for ROV multibeam sonar observations from 24 June to 28 July 2020; **(c)** surface photography with the investigated ice ridge internal sail structure on 4 July 2020; **(d)** water temperature and water freezing temperature from microstructure profiles (MSS) at 5-m depth; **(e)** structure of a sea-ice ridge. Displayed ice edges in (b) were derived from AMSR-2 sea-ice concentration (SIC) product for thresholds of 15 % and 80 % on 28 July 2020 (Spreen et al., 2008).

## 2.1 Ridge drilling

In this study, we focus primarily on the evolution of a ridge called 'Jaridge'. Jaridge was formed between 4–12 February 2020, based on the visual inspection of sea-ice surface elevation models from an airborne laser scanner (Jutila et al., 2022). The ice blocks forming the ridge were 0.2–0.4 m thick (Fig. 1c), the average sail height was 0.5 m, and the average draft was 3.8 m. It was formed between level first-year ice and level second-year ice. We investigated ridge morphology using a 2-inch diameter ice auger (Kovacs Enterprise, USA) and thickness tape to measure snow or ice interface position. Ice drilling was organized along seven drilling transects perpendicular to the ridge crest orientation (Fig. 2a). Each transect contained 3–7 drilling locations with measurements of ice draft, freeboard, depth of ridge voids, and snow thickness at a horizontal spacing of 2.5 or 5 m (Fig. 2b). The ridge was measured seven times (25 June to 29 July) with a total of 47 drill holes during the summer melt season when located over the Yermak Plateau and Fram Strait (79.4–82.1° N, 2.8° W–10.2° E, Fig. 1b). Jaridge covered 12 % of the area of sonar surveys, which included four classified ice types (Fig. 2c). Another shallower ridge, 'Porridge', was also located within the survey area but only mapped with the multibeam sonar. The area at the top right quarter of sonar surveys was heavily covered with false bottoms during 7–29 July (Salganik et al., 2023b) and was therefore excluded from our analysis.

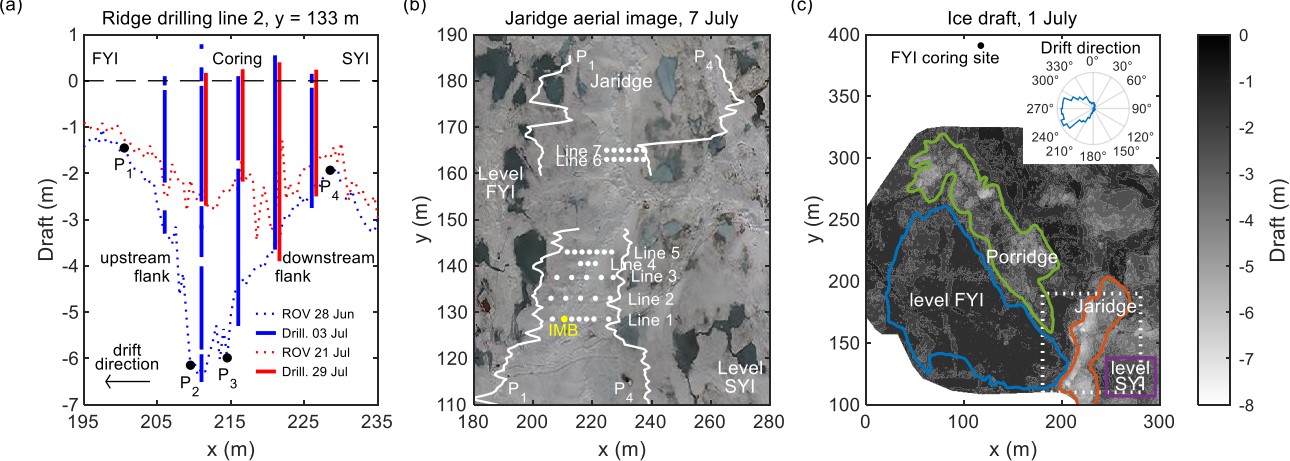

**Figure 2: (a)** Cross-section of ice draft in late June and late July 2020 along drilling line 2. The vertical lines are drill holes, the solid lines are ice, and the line breaks are voids, corners $P_{1-4}$ represent trapezoidal ridge shape; **(b)** locations of ridge drillings, ice mass balance buoy (IMB) and keel width boundaries $P_1$ and $P_4$ of Jaridge on an optical helicopter-borne aerial image from 7 July; **(c)** ice bottom topography on 1 July 2020, measured by remotely operated vehicle (ROV) multibeam sonar, showing location of first-year ice (FYI), second-year ice (SYI), Jaridge and Porridge and location of (b) inside white dotted-line box. The polar histogram in (c) shows frequency of ice drift direction in relation to the displayed ice floe orientation, with prevailing drift in western direction.

To study the temporal evolution of the ridge interfaces, we used temperature measurements from IMB 2020M26 (Bruncin d.o.o.). The IMB consisted of a 5-m-long thermistor chain with a sensor spacing of 2 cm and provided temperature readings

every 6 hours with an accuracy of 0.1°C and daily heating-induced temperature difference measurements after a cycle of internal heating, allowing to identify the location of snow-ice and ice-water interfaces with high precision (Jackson et al., 2013). The IMB was installed on 26 June 2020, at the ridge drilling Line 1 (Fig. 2b). At the IMB site upon deployment, the consolidated layer thickness was 1.9 m, the keel draft was 4.0 m, and the snow depth was 0.6 m. Temperature, salinity, and isotope compositions from Jaridge coring are presented in Lange et al. (2023), with ridge bulk salinity of 1.8–2.8 and snow mass fraction of 6–11 %.

## 2.2 Underwater multibeam sonar

We use a multibeam sonar (DT101, Imagenex, Canada) mounted on a remotely operated vehicle (ROV, M500, Ocean Modules, Sweden, after Katlein et al. (2017)) to measure ice draft in an area of approximately 350 m by 200 m, with 0.05 m vertical accuracy and 0.5 m horizontal resolution. The surveys were done in a grid pattern with a distance between lines of 20–25 m. The sonar had 480 beams with an across track swath width of 120° (64 m width for level ice), an along track swath width of 3°, an effective beam width of 0.75°, and an angular resolution of 0.25°. Seven surveys at a depth of 20 m were performed during the melt season (24 June to 28 July), close to the floe edge of the Central Observatory of MOSAiC (Nicolaus et al., 2022), covering an area with undeformed ice and several ice ridges, including the Jaridge (Fig. 1b). In our analysis, we mostly use the first six sonar surveys, as ice deformations decreased the co-location accuracy of sea ice ridges for the last survey.

## 2.3 Ridge morphology analysis

To quantify how ridge characteristics affect melt rates, we divided our ridge draft multibeam observations into 131 individual cross-sections that were nearly parallel to the direction of ice drift during June–July. The distance between neighbouring cross-sections was 0.5 m. We determined the following characteristics for each cross-section: keel bottom width, draft, slope, and distance from the keel front. To quantify these parameters with a single value, we simplified each cross-section to a trapezoidal shape, following Ekeberg et al. (2015). Four points of these trapezoids ($P_1$–$P_4$, Fig. 2a) coincide with the largest transition of the smoothed inclination of ridge cross-sections, separating each cross-section into upstream flank, keel bottom, and downstream flank (locations of $P_1$ and $P_4$ are shown in Fig. 2b). The upstream flank was facing the ice drift direction, while the downstream flank was on the leeside of the prevailing ocean current relative to the ice (Fig. 2c). The keel bottom width is equal to the horizontal projection of the keel bottom ($P_2$–$P_3$), while the keel draft is equal to the average draft of the keel bottom. The keel slope is defined as the angle between the upstream flank and the waterline. A tangent straight line "touching" the initial position of all $P_2$ points of cross-sections (upstream bottom corners) on June 24 is the keel front (Fig. 4c). The distance from $P_2$ of each cross-section to the keel front was identified as one of the factors for studying ridge melt rates.

**2.4 Relationship between sea ice draft and thickness**

This study is focused on the measurements of sea ice draft, while the draft evolution may provide only an estimate of sea ice melt as it involves several complex processes affecting the parameters required for such conversion. Under the assumption of hydrostatic equilibrium, the sea-ice draft decrease equals the thickness of surface and bottom melt multiplied by snow and sea-ice density, respectively, and divided by water density (Fons et al., 2023). For various remote sensing measurements, including satellite altimetry (Landy et al., 2022) and upward-looking sonars (Sumata et al., 2023), the sea ice density is assumed to be

constant, while considering sea-ice density seasonal evolution may improve the accuracy of satellite ice thickness retrievals (Fons et al., 2023). In sections 3.1–3.4, we compare raw measurements of draft change and estimates of sea ice melt under the assumption of a constant draft to thickness ratio of 0.9.

We used the ridge drilling described in section 2.1 to study the relationship between ridge draft, freeboard, snow thickness, and macroporosity. Additionally, we performed sea-ice density measurements from cores extracted from Jaridge as well as

from the other ridge within the MOSAiC floe. To study the evolution of level ice draft, thickness, and interface evolution, we used data from the first-year ice (FYI) coring site located 70 m away from the ridge surveys (Fig. 2c), further detailed in Salganik et al. (2023b). These observations include a combination of IMB temperature measurements and sea-ice coring conducted on a weekly basis, with measurements of FYI temperature, salinity, and density, as well as snow and ice thickness and draft from 20–30 sea-ice cores per week. In sections 3.5–3.6, we present our measurements of sea-ice density and draft to

thickness ratio evolution and show how these measurements can refine the estimates of sea-ice melt for different ice types. In our sea-ice density estimates, the gas fraction was calculated from laboratory hydrostatic measurements of sea-ice density with an error below 2 % (Pustogvar and Kulyakhtin, 2016), while brine volume was calculated from in situ temperature and salinity measurements (Cox and Weeks, 1983).

**3 Results and discussion**

**3.1 Level ice melt**

In this study, we focus on the observed difference in sea-ice draft between the sonar surveys from 24 June to 21 July due to the large spatial variability in melt rates. During this period, an area of undeformed FYI (Fig. 2c) with an initial draft of 1.4±0.2 m experienced a 0.42±0.26 m decrease in draft, while an area of undeformed SYI with an initial draft of 2.6±0.7 m decreased by 0.50±0.31 m (20 % more than FYI). A shallow ridge ('Porridge') with an initial draft of 2.3±0.8 m (similar to

SYI) experienced a 0.54±0.61 m decrease in draft. FYI draft decrease had a positive correlation with its initial draft, with a regression slope of 0.47. Such a relationship may be related to the strong vertical stratification of the ocean mixed layer observed in July (Fer et al., 2022). Skyllingstad et al. (2003) suggested that fresh water insulates sea ice if turbulent mixing is weak, while thicker ice and ridges are efficient at forcing turbulence in the fresh layer. This agrees with our measurements of

FYI, SYI, and ridge bottom temperatures (Fig. 1a), where thinner ice was more strongly affected by meltwater for a longer
time, which led to a lower OHF.

## 3.2 Ridge morphology and keel melt

Repeated ridge drilling showed that Jaridge keel melt was very variable (Fig. A1). The average melt along ridge drilling lines
1–3 (Fig. 2b) was 1.7 m, while ridge flanks melted up to 4.5 m. For the ROV sonar surveys, the average draft change of the
ridge area was 0.9±1.0 m with an average initial draft of 3.9±1.1 m (Fig. 3a). The maximum ridge draft decreased from 8.2 m
to 7.0 m, while the largest observed ridge draft reduction was 5.6 m (not including area with mechanical erosion). The average
keel slope was 14–15° for both flanks, half of that reported by Wadhams & Toberg (2012), possibly because of the larger 5 m
minimum ridge draft threshold used in their study. The average initial fraction of keel bottom width ($P_2$–$P_3$, Fig. 2a) to keel
width ($P_1$–$P_4$) was 38 %, twice as large as the 17 % estimated by Ekeberg et al. (2015), possibly related to the larger maximum
cross-sectional keel bottom draft (7.2 m in comparison to our 5.3 m) and 5 m keel draft threshold. Co-location of ridge draft
measurements from drilling and from sonar showed good agreement between the two draft measurement techniques ($R^2 = 0.8$,
Fig. A1). According to the individual observations of ice draft evolution from multibeam sonar, the melt of ridge flanks
stronger (1.7 times larger regression slope) depended on ice draft in comparison to the melt of keel bottom (Fig. 3b). The
average melt at the same depth was higher for flanks than for keel bottom. For example, for an ice draft larger than 4 m, the
average draft change for upstream flank, keel bottom, and downstream flank was 1.3 m, 1.0 m, and 1.4 m, respectively. Fig.
3b can be used to predict the ridge melt relative to level ice melt depending only on ridge draft and fraction of keel bottom
width. On average, ridge flanks and keel bottom were melting 1.7 and 2.0 times faster than FYI at the coring site. Higher
average melt rate for keel bottom was related to higher average initial draft for keel bottom (4.4 m) than for flanks (3.1 m).

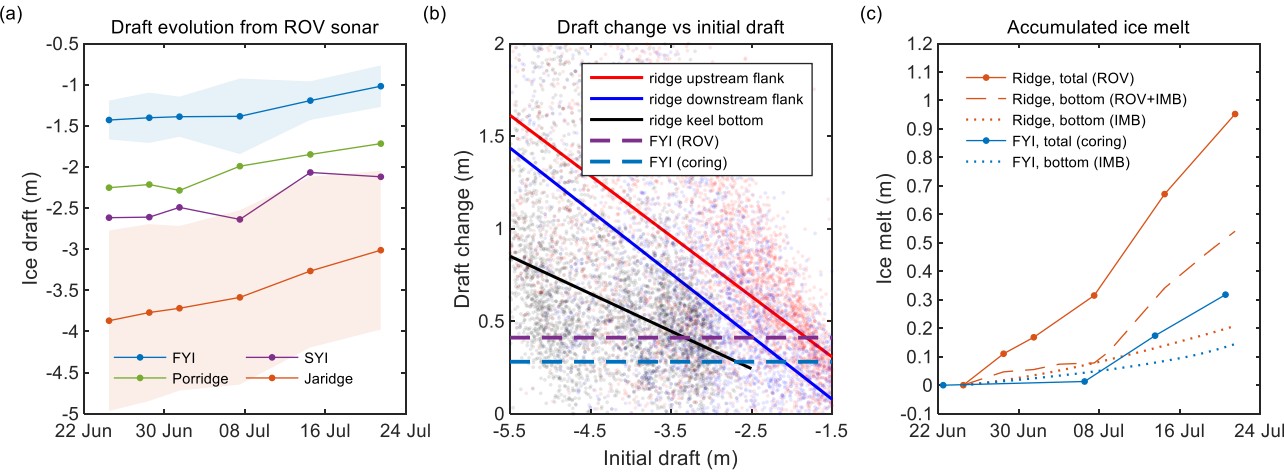

**Figure 3: (a) Evolution of the average sea-ice draft measured by a ROV multibeam sonar for first-year ice (FYI), second-year ice**
**(SYI), Porridge and Jaridge during June–July 2020; (b) draft change for single-point sonar measurements of ridge upstream and**
**downstream flanks and keel bottom, corresponding linear regression with solid lines, and average draft change for FYI coring site;**
**(c) accumulated ice melt for ridge and FYI estimated from ROV multibeam sonar, ice mass balance buoy (IMB) and ice coring**
**measurements. Shaded areas in (a) represent standard deviations of draft measurements.**

### 3.3 Ridge cross-sectional melt

Based on the results of multiple linear regression analysis, keel draft, slope, bottom width, and distance to the keel front (Fig. 4c) are responsible for 57 % (coefficient of determination $R^2$) of ridge melt variability, with 37 % positive correlation with keel draft, 32 % positive correlation with keel slope, 27 % negative correlation with keel bottom width, and 11 % negative correlation with distance from the keel front. The roughness of the ridge keel, characterized by its draft standard deviation, did not have a significant effect on the ridge melt. The large correlation of ridge melt with its draft may be explained by a

combination of both higher ice melt and lower keel width for larger drafts. Based on ridge drilling observations from this study and from another examined ridge during MOSAiC (Salganik et al., 2023a), the flanks of ridge keels are usually less consolidated, which may be coupled with higher ocean turbulence at the ridge flanks in comparison to the keel bottom.

The keel bottom width ranged from 1 m to 35 m with an average of $13\pm9$ m, and the highest melt was observed around the ridge left and right bottom corners ($P_2$ and $P_3$) within a diameter of 10 m (Fig. 4a). For wide profiles, it was possible to

distinguish keel melt around two bottom corners and in the keel bottom between them. While areas within 10 m around the upstream bottom corner ($P_2$) melted on average by 1.2 m, keel bottom without 10 m surroundings around both corners ($P_2$ and $P_3$) melted by 0.5 m (similar to level ice melt rates despite a much larger ice draft). We also found that all ridge cross-sections that had both narrow keel bottom width and low keel melt were located within two areas (Fig. 4c) and were characterized by a large distance from the keel front (their upstream flanks were shifted towards the ridge interior). Exclusion of profiles from

these two areas would increase the correlation ($R^2$) between keel melt and keel bottom width from 27 % to 57 % alone (Fig. 4b). We suggest that these areas were protected by the keel front from the turbulent fluxes, which appear to occur in the vicinity of the ridge bottom corners ($P_2$ and $P_3$ in Fig. 2a). The ridge at y = 148–160 m was not trapezoidal and consisted of separate blocks with patchy draft evolution. This area with mechanical erosion was not included in the correlation analysis but was included in the further comparison of level ice and ridge melt. The mean draft decrease for this area was $1.0\pm2.1$ m, 10 %

higher than for the rest of the ridge. The draft decrease of other ridge parts was gradual, and they were fully consolidated, suggesting little mechanical erosion. The strong negative correlation between keel melt and keel bottom width may be explained by the stronger ridge consolidation at its interior parts (Salganik et al., 2023a; Shestov et al., 2018), as well as by the smaller fraction of the keel affected by the enhanced turbulence around the keel bottom corners $P_2$ and $P_3$.

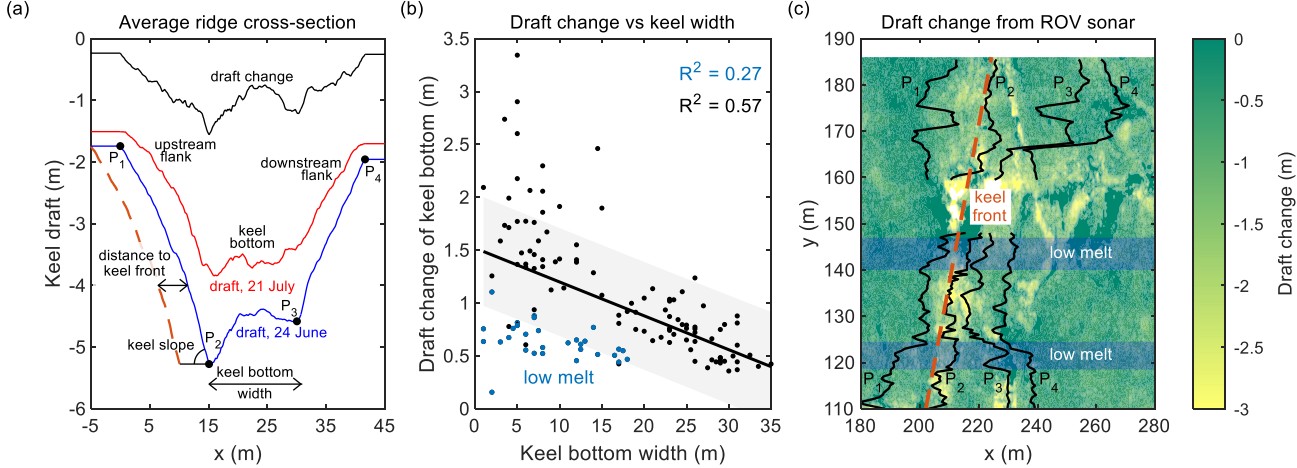

**Figure 4: (a) Average ridge cross-section of ice draft in late June and late July 2020; (b) draft change of keel bottom vs keel bottom width for each ridge cross-section; (c) contour plot of ridge draft change from 24 June to 21 July with locations of ridge corners P₁₋₄ (black lines), keel front (red dashed line), and cross-sections with low total melt and narrow keel width (blue shaded areas).**

## 3.4 Total, surface, and bottom ice melt

In the previous sections, we analysed the draft evolution of several sea ice types. Meanwhile, it is important to separate surface and bottom melt to study the thermodynamic coupling of sea ice, ocean, and atmosphere. In this section, we assume a constant ratio of sea-ice draft and thickness of 0.9, characteristic of unchanging sea-ice density. From 22 June to 20 July, unponded level ice at the FYI coring site experienced 0.08 m snow melt, 0.18 m surface melt, and 0.14 m bottom melt, with nearly identical draft change (0.34 m) and total melt (0.32 m). Meanwhile, sonar measurements gave a larger FYI draft change (0.41 m), and hence provided a substantially larger estimate of FYI bottom melt (0.25 m) under the assumption of the same snow and ice surface melt as for the FYI coring site. The reason of such difference is discussed in section 3.6.

During the same period, the average snow depth (for each drilling survey) above Jaridge decreased from 0.50 m to 0.12 m. Temperature and heating-induced temperature difference measurements from IMB indicate a surface ridge melt of 0.24 m. Macfarlane et al. (2021) present an average snow density of 420 kg/m³ in June and July. Assuming 0.24 m of surface melt and 0.38 m of snowmelt for the whole ridge, using sonar measurements, we can estimate the average ridge bottom melt as 0.55 m, or 60 % of the mean ridge total melt of 0.93 m. This may explain why only 57 % of the ridge total melt was related to characteristics of the keel topography. The surface melt of level FYI and the ridge was similar, whereas the ridge bottom melt estimates were 2.2–3.9 times larger than for level FYI.

The average ridge macroporosity measured by drilling in June–July was 4±7 % for all 47 drilling sites (Fig. A1). Bottom ridge brine volume (5–7 %) was lower than for FYI due to lower ridge temperatures (Fig. 1a). This shows that the ridge macroporosity and brine volume fraction have a minor effect on the estimate of the total volume of melted ice based on its draft measurements relative to the difference in melt between various ice types. A study focused on the seasonal evolution of ridge consolidation based on observations from IMBs and ice drilling during MOSAiC showed that the most consolidation

occurred during the spring season, while upon melt onset, ridges were already fully consolidated (Salganik et al., 2023a).
Following Lei et al. (2022), we used 10 MOSAiC IMBs to estimate an average bottom melt of 0.17±0.07 m for first- and
second-year level ice with an average initial thickness of 1.8±0.2 m from 22 June to 20 July, with no significant correlation
between the initial ice thickness and bottom melt.

## 3.5 Effect of meltwater drainage on ice draft

From 7 July to 14 July, we observed unexpected 0.08 m increase in FYI freeboard at the coring site, despite 0.16 m total melt
(Fig. 5a). We suggest that this short-term imbalance is related to surface melt pond drainage observed from 9 July to 13 July
(Webster et al., 2022), which was accompanied by the formation of an under-ice meltwater layer with 21 % areal coverage
and 0.46 m thickness (Salganik et al., 2023b). This suggests that the large decrease in draft (0.30 m) for FYI measured by
sonar during 7–14 July was not purely due to ice melt but includes an approximately 0.10 m freeboard increase (Fig. 5d).
During that period, independent measurements from FYI coring also showed a substantially larger draft decrease (0.24 m in
comparison to 0.08 m draft change during 14–21 July). Meanwhile, the total FYI melt from coring during these two weeks
was 0.16 m and 0.14 m, respectively. Our observations also indicate that the increase of FYI freeboard caused by meltwater
drainage was reversible (Fig. 5a). It is supported by the rapid recovery of surface melt pond fraction and depth to the values
prior to drainage during 13–17 July (Webster et al., 2022).

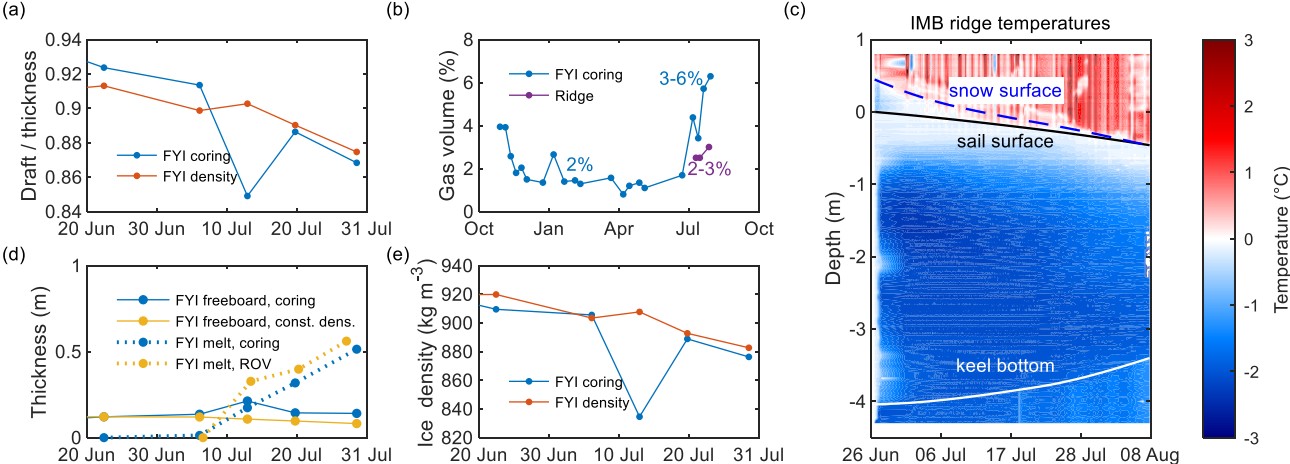

**Figure 5: (a) Evolution of first-year ice (FYI) draft and thickness ratio; (b) FYI and ridge gas volume; (c) contour plot of the ridge**
**temperature measurements from IMB buoy 2020M26 with sail surface and keel bottom interfaces from IMB and average snow**
**thickness from the ridge drilling surveys; (d) FYI freeboard and total melt; (e) FYI density estimated from coring measurements of**
**snow and ice thickness and draft, and from density measurements.**

## 3.6 Effect of sea-ice density on ice draft

Previously, we used a constant ratio of sea-ice draft and thickness to estimate its melt from the draft measurements. Meanwhile,
at the coring site, the ratio of FYI draft to thickness gradually decreased from 0.92 on 22 June to 0.87 on 29 July (Fig. 5a). The

corresponding estimate of sea-ice bulk density (assuming hydrostatic equilibrium) decreased from 910 kg/m$^3$ to 876 kg/m$^3$, which agrees with a sea-ice density decrease from 914 kg/m$^3$ to 875 kg/m$^3$ from the direct density measurements performed at the FYI coring site (Fig. 5e). During these measurements, the snow at the FYI coring site melted entirely by 20 July from an initial depth of 0.08 m and had a minor effect on FYI freeboard. The observed sea-ice density decrease was mainly caused by an increase in gas fraction from 2 % to 6 % (Fig. 5b). We suggest that the decrease of FYI density led to a large difference in FYI total melt estimates from coring thickness measurements (0.34 m) and from sonar draft measurements (0.46 m), indicating that the FYI melt from sonar may be overestimated when assuming a constant sea-ice density (Fig. 5d) or draft to thickness ratio.

Unlike for the level ice, the ratio of draft to thickness for Jaridge drilling lines was 0.89±0.06 and did not decrease (Fig. A1), while the ridge bulk density estimated from coring measurements on 10 July was 892 kg/m$^3$. The ridge gas volume fraction was 2.5–3.0 % (Fig. 5b), while in contrast to the level ice, the ridge was also colder than seawater (Lange et al., 2023). A strong decrease in sea-ice density was not observed at the SYI coring site, with a nearly constant gas fraction during melt season (Salganik et al., 2023b). These measurements suggest that considering the draft-to-thickness ratio dependence on the sea-ice density evolution of different ice types may improve estimates of ice melt from sonar surveys. Therefore, we suggest 3.8 times higher bottom and 3.0 times higher total melt rates for the sea-ice ridge than for FYI. For a typical areal fraction of sea-ice ridges (40–50 %), we estimate that they produce 1.7–2.5 times more meltwater than level ice.

The absence of ridge lift during melt season is supported by sonar measurements with a smaller draft change of FYI and SYI (0.24–0.25 m) right next to the ridge in comparison to the average FYI and SYI draft change of 0.41–0.50 m away from the ridge. Measurements from an airborne laser scanner (0.025 m accuracy, Ricker et al., 2023) give 0.02 m increase in FYI freeboard during 4–17 July, which agrees with 0.01 m freeboard increase from FYI coring during 6–20 July, supporting our density measurements and upscaling them for the whole FYI area, surveyed by multibeam sonar. Our FYI coring measurements also agree with previous estimates of sea-ice density seasonal evolution (Fons et al., 2023). A gradual increase of FYI freeboard from 22 June to 29 July by 0.02 m despite a total FYI melt of 0.52 m, observed at the FYI coring site and mainly caused by the decrease of FYI density, may affect aerial and satellite altimetry retrievals in the Arctic summer.

Measurements of sea-ice bottom melt allow us to estimate the OHF for different ice types, following Shestov et al. (2018). From 24 June to 21 July, calculations based on temperature measurements from the FYI IMB resulted in an average OHF of 17 W m$^{-2}$, increasing from 11 W m$^{-2}$ to 36 W m$^{-2}$ (Salganik et al., 2023b). A combination of sonar and IMB measurements at the ridge result in an average OHF of 65 W m$^{-2}$ with averages of 20 W m$^{-2}$ during 24 June – 7 July and 107 W m$^{-2}$ during 8 July – 21 July, respectively, suggesting larger ridge melt enhancement for lower sea-ice concentrations of 85 % (Krumpen et al., 2021) due to increased solar heat input.

**3.7 Comparison with previous observations of enhanced ridge melt**

The areas representative for the Arctic sea ice cover are characterized by high ice concentration and comparable surface and bottom level ice melt, while the largest amount of bottom level ice melt occurs in regions with low ice concentration (Perovich

et al., 2011). The average accumulated surface/bottom melt of level ice was 0.56/0.50 m for yearlong SHEBA measurements and 0.24/0.31 m for MOSAiC, with measurements until 29 July, covering approximately half of the SHEBA melt season. The average OHF was 18 W m$^{-2}$ from 3 June to 4 October for SHEBA and 18 W m$^{-2}$ from 3 June to 29 July for MOSAiC FYI coring site, similar to the average summer estimates for Beaufort Gyre and Transpolar Drift (Krishfield, 2005). This suggests that level ice melt during MOSAiC was comparable to SHEBA, with a surface and bottom melt ratio typical for areas with high ice concentration. Despite a deeper ridge keel with a 6 m total ice thickness on SHEBA, the enhanced ridge bottom melt relative to level ice was only 60 % (Perovich et al., 2003) in comparison to 280 % for MOSAiC with average keel draft of 3.9 m and sail height of 0.5 m. This difference could be related to the older (second-year and multiyear) SHEBA ridges or to the substantially smaller areal coverage of SHEBA ridge measurements. Second- and multiyear ice ridges surveyed by Perovich et al. (2003) are typically smooth, fully consolidated, and have low sea-ice salinity, as observed previously by Kovacs et al. (1973); and this may affect turbulence around old ridges. With only 14 stakes at the second-year ridge during SHEBA, in comparison to over 10$^4$ ridge draft measurements for MOSAiC, the latter captures the whole range of different melt rates.

The oceanographic conditions during N-ICE2015 were substantially different from both SHEBA and MOSAiC due to the proximity to Atlantic Water, with the average OHF under level ice of 63 W m$^{-2}$ during 10–19 June 2015 (Peterson et al., 2017), six times higher than for MOSAiC during the same period. The enhanced OHF for ice ridges during N-ICE2015, observed by Shestov et al. (2018), is based on one single-point measurement of bottom ridge melt from a temperature buoy and an OHF estimate from turbulence instrument clusters for level ice. The keel macroporosity of the ridge studied by Shestov et al. (2018) was 8 % (higher than 4 % for MOSAiC, possibly due to the lower N-ICE2015 keel width of 16 m), the maximum keel draft was 7.3 m, and the estimated increase of the ridge bottom melt in comparison to level ice was 4.8 compared to 3.8 for MOSAiC. The enhanced first-year ridge melt estimated from draft measurements from upward-looking sonar in Beaufort Sea from Amundrud et al. (2006) may have substantial uncertainties as they are based on an assumption of similar ice draft distribution along the direction of ice drift (and not necessarily repeated measurements of the same ridges), while ridge macroporosity, block thickness, sea-ice density, and fraction of surface and bottom ice melt were unknown. Amundrud et al. (2006) estimated the total level ice melt rate in July as 0.02–0.03 m d$^{-1}$, similar to 0.018 m d$^{-1}$ for the MOSAiC FYI coring site (1–29 July), while the total melt rate for ridges with 4–8 m draft was 0.10 m d$^{-1}$ in the Beaufort Sea and 0.04 m d$^{-1}$ during MOSAiC (1–21 July). The corresponding ridge total melt enhancement of 4.0 was higher than the ratio of ridge and level ice total melt of 3.0 for MOSAiC. Another ridge (Alli's Ridge) with similar block thickness, keel draft, width, and macroporosity but oriented along the drift direction (perpendicular to Jaridge) was studied during MOSAiC. Despite the different orientation, by 26 July Alli's ridge experienced a similar draft decrease of 0.9 m as Jaridge; however, those observations are limited to four point measurements across a single ridge cross-section.

The results of our sonar investigations provided evidence of high spatial variability in sea ice melt, especially for ridges. For the ridge with a total cross-sectional melt of 0.9±0.4 m, we identified cross-sections with an average total melt ranging from 0.2 m to 2.6 m. This means that measurements from a single location or even a single ridge cross-section may not be representative, as the variability of draft change is comparable to the difference between the average melt of different sea ice

types. This suggests that the 20–30 % difference between our observations of first-year ridge enhanced melt and observations from Amundrud et al. (2006) and Shestov et al. (2018) may be related to the high spatial variability of ridge melt or different
oceanographic conditions, while the much larger 120 % difference from Perovich et al. (2003) may be attributed to different ridge age.

### 3.8 Study application and limitations

Our study provides melt estimates for different ice types over a substantial range of sea ice draft. It also describes the effects of ridge morphological parameters, including ridge draft, slope, and width, on melt rates. This allows us to extrapolate our
estimates to a range of ridge cross-sections with various shapes and drafts. Nevertheless, we acknowledge limitations related to some other ridge characteristics that may affect melt, such as macroporosity, block thickness, and ridge age. Most of the characteristics of the investigated ridge are close to the average characteristics of first-year ice ridges with 8 m maximum keel draft, 36 m keel width, and 0.2–0.4 m block thickness (Strüb-Klein and Høyland, 2011), except for its low macroporosity (the number of ridges sampled during the melt season is limited, but melting ridges often have low macroporosity (Marchenko,
2022; Shestov et al., 2018)). Similarly, our results do not cover ridges with keel drafts above 8 m and strongly non-trapezoidal shapes, often observed in Arctic regions with thicker sea ice, including the Chukchi and Beaufort Seas (Metzger et al., 2021). Previous studies on ridge melt showed a wide range of ridge and level ice melt fractions from only 1.6 (Perovich et al., 2003) to 4–5 (Amundrud et al., 2006; Shestov et al., 2018), yet due to the limited coverage of these observations, it is challenging to indicate which parameters were the main reason for such differences in ridge melt. Additional limitations are related to the
estimates of sea-ice density seasonal evolution for different ice types due to the small number of such observations, especially for ridges, and known challenges and uncertainties related to brine losses during ice coring (Pustogvar and Kulyakhtin, 2016).

### 4 Conclusions

We collected an unprecedented dataset using a multibeam sonar mounted on an ROV that captured the four-dimensional change of sea-ice draft over a period of one month during advanced summer melt in the Arctic Ocean. This revealed that a
390 first-year ridge melted faster than adjacent level ice types. The total ridge melt was on average 0.95 m, compared to 0.55 m for level second-year ice and 0.46 m for level first-year ice. These observations can largely be explained by the difference in initial average ice draft of 1.4 m for first-year ice, 2.6 m for second-year ice, and 3.9 m for the ridge. Ridge bottom melt was 3–4 times higher than the bottom melt of first-year level ice, while surface melt was almost identical. The high-resolution sonar observations also revealed large spatial heterogeneity in keel melt, and therefore the results from point observations need
to be interpreted with care since it is difficult to tell how representative they are.

Key factors that affect the melt rates of ridge keels included the keel draft and slope (with negative correlation), keel width and distance from the keel front (with positive correlation). These factors can explain 57 % of the total melt variability for this particular ridge, with 36 % of the melt variability explained by keel draft, 32 % by keel slope, 27 % by keel width, and 11 %

by a distance from the keel front. We observed a relationship between the melt of ridge flanks with their draft, and amplification of keel melt within 10 m of its bottom corners, while melt rates of the (more level) middle part of ridge keel bottom were comparable to level ice melt. However, ice draft changes (as measured by sonars) are not due to ice melt alone, because the hydrostatic balance of the ice needs to be considered, since, e.g., melt pond drainage and sea-ice density evolution change ice draft. This needs to be considered when such measurements are used over longer periods of time. Considering the seasonal change in sea-ice density allowed us to refine the ratio of total ridge to first-year level ice melt to 3.0 and the ratio of bottom ice melt to 3.8. Such ice draft changes also affect the ice freeboard and can potentially affect satellite altimetry retrievals in the Arctic summer.

Since a large fraction of the Arctic ice pack is deformed (ridged) ice, it is imperative that we better understand their role in the Arctic sea ice system. While ridge keels contribute a significant amount of ice melt in summer (Perovich et al., 2021), they also provide a sink for meltwater through refreezing in keel voids (Lange et al., 2023). Ridge keels also shape the lateral distribution of under-ice meltwater layers (which in turn affect level ice melt rates) (Salganik et al., 2023b) and affect turbulent exchanges (Skyllingstad et al., 2003), with implications for ice-ocean exchange. This work highlights areas that warrant future observation-model development for improved representation of ridge-related sea-ice processes in models.

## Appendix A: Additional information

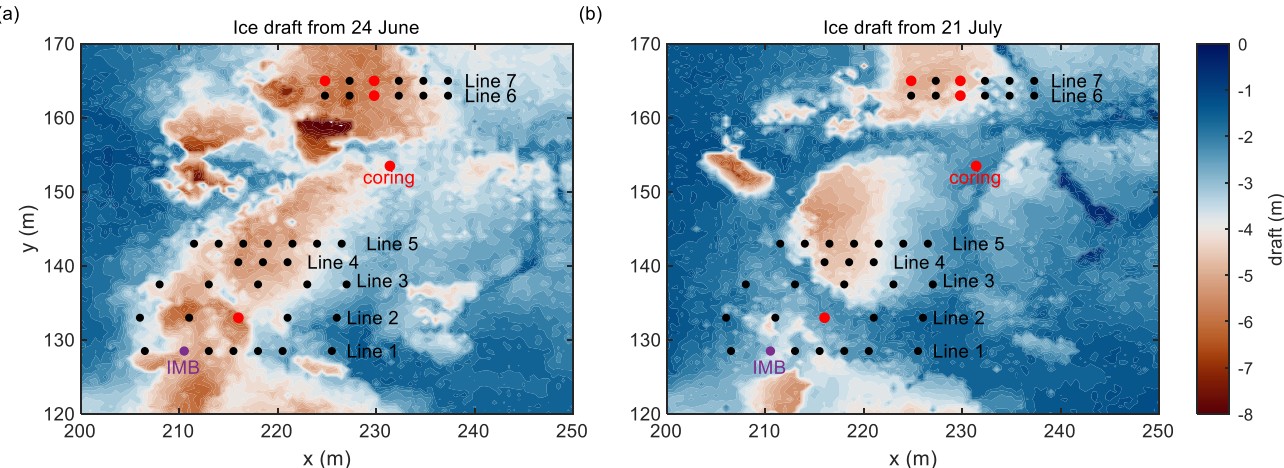

(a) Line 1 (y = 129 m)
(b) Line 3 (y = 138 m)
(c) Line 4 (y = 141 m)
(d) Line 5 (y = 143 m)
(e) Line 6 (y = 163 m)
(f) Line 7 (y = 165 m)

**Figure A1: Ridge draft measurements from ROV multibeam sonar and ice drilling lines. The vertical lines are drill holes, the solid lines are ice, and the line breaks are voids, the dotted lines are ROV sonar draft measurements.**

(a) Ice draft from 24 June
(b) Ice draft from 21 July

**Figure A2: Contour plot with ice draft for 24 June (a) and 21 July (b). Black points show ridge drilling locations, purple point shows the location of ice mass balance buoy (IMB), and red points show the location of ridge coring. Contour plot colours follow recommendations of scientifically derived colour maps (Crameri et al., 2020).**

**Data availability**

All scientific data used in this study is publicly available:

Granskog, M. A.; Lange, B. A., Salganik, E., De La Torre, P. R., Riemann-Campe, K.: Temperature and heating induced temperature difference measurements from the modular buoy 2020M26, deployed during MOSAiC 2019/20. *PANGAEA*, https://doi.org/10.1594/PANGAEA.938354, 2021.

Jutila, A., Hendricks, S., Birnbaum, G., von Albedyll, L., Ricker, R., Helm, V., Hutter, N., Haas, C.: Geolocated sea-ice or snow surface elevation point cloud segments from helicopter-borne laser scanner during the MOSAiC expedition, version 1. *PANGAEA*, https://doi.pangaea.de/10.1594/PANGAEA.950509, 2022.

Katlein, C., Anhaus P., Arndt S., Krampe, D., Lange, B. A., Matero, I., Regnery, J., Rohde, J., Schiller, M., Nicolaus, M.: Sea-ice draft during the MOSAiC expedition 2019/20. *PANGAEA*, https://doi.org/10.1594/PANGAEA.945846, 2022.

Lange, B. A., Salganik, E., Macfarlane, A. R., Schneebeli, M., Høyland, K. V., Gardner, J., Müller, O., Granskog, M. A.: Ridge ice oxygen and hydrogen isotope data MOSAiC Leg 4 (PS122/4). *PANGAEA*, https://doi.org/10.1594/PANGAEA.943746, 2022.

Macfarlane, A. R., Schneebeli, M., Dadic, R., Wagner, D. N., Arndt, S., Clemens-Sewall, D., Hämmerle, S., Hannula, H.-R., Jaggi, M., Kolabutin, N., Krampe, D., Lehning, M., Matero, I., Nicolaus, M., Oggier, M., Pirazzini, R., Polashenski, C., Raphael, I., Regnery, J., Shimanchuck, E., Smith, M. M., Tavri, A. Snowpit raw data collected during the MOSAiC expedition. PANGAEA, https://doi.org/10.1594/PANGAEA.935934, 2021.

Neckel, N., Fuchs, N., Birnbaum, G., Hutter, N., Jutila, A., Buth, L., von Albedyll, L., Ricker, R., Haas, C.: Helicopter-borne RGB orthomosaics and photogrammetric Digital Elevation Models from the MOSAiC Expedition. *PANGAEA*, https://doi.pangaea.de/10.1594/PANGAEA.949433, 2022.

Oggier, M., Salganik, E., Whitmore, L., Fong, A. A., Hoppe, C. J. M., Rember, R., Høyland, K. V., Divine, D. V., Fons, S. W., Abrahamsson, K., Aguilar-Islas, A. M., Angelopoulos, M., Balmonte, J. P., Bozzato, D., Bowman, J. S., Chamberlain, E., Creamean, J., D'Angelo, A., Gardner, J., Haapala, J., Immerz, A., Kolabutin, N., Lange, B. A., Lei, R., Marsay, C. M., Maus, S., Olsen, L. M., Müller, O., Ren, J., Rinke, A., Sheikin, I., Shimanchuk, E., Spahic, S., Torres-Valdés, S., Torstensson, A., Ulfsbo, A., Wang, L., Granskog, M. A.: First-year sea-ice salinity, temperature, density, oxygen and hydrogen isotope composition from the main coring site (MCS-FYI) during MOSAiC legs 1 to 4 in 2019/2020. PANGAEA, https://doi.pangaea.de/10.1594/PANGAEA.956732, 2023.

Salganik, E., Lange, B. A., Sheikin, I., Høyland, K. V., Granskog, M. A. (2023). Drill-hole ridge ice and snow thickness and draft measurements of "Jaridge" during MOSAiC 2019/20. *PANGAEA*, https://doi.org/10.1594/PANGAEA.953880

Salganik, E., Lange, B. A., Høyland, K. V., Gardner, J., Müller, O., Tavri, A., Mahmud, M. Granskog, M.A.: Ridge ice density data MOSAiC Leg 4 (PS122/4). PANGAEA, https://doi.org/10.1594/PANGAEA.953865, 2023.

Schmithüsen, H. 2021. Continuous meteorological surface measurement during POLARSTERN cruise PS122/4. Alfred Wegener Institute, Helmholtz Centre for Polar and Marine Research, Bremerhaven. PANGAEA, https://doi.org/10.1594/PANGAEA.935224.

Schulz, K., Mohrholz, V., Fer, I., Janout, M. A., Hoppmann, M., Schaffer, J., Koenig, Z., Rabe, B., Heuzé, C., Regnery, J., Allerholt, J., Fang, Y.-C., He, H., Kanzow, T., Karam, S., Kuznetsov, I. Kong, B., Liu, H., Muilwijk, M., Schuffenhauer, I., Sukhikh, N., Sundfjord, A., Tippenhauer, S.: Turbulent microstructure profile (MSS) measurements from the MOSAiC drift, Arctic Ocean. *PANGAEA*, https://doi.org/10.1594/PANGAEA.939816, 2022.

Schulz, K., Koenig, Z., Muilwijk, M.: The Eurasian Arctic Ocean along the MOSAiC drift (2019-2020): Core hydrographic

parameters. Arctic Data Center, https://doi.org/10.18739/A21J9790B, 2023.

**Author contribution**

ES, BAL, CK, IM, KVH and MAG contributed to the design of the study. ES, BAL, CK, IM and MM collected and processed the field data. ES undertook the statistical analyses and interpreted the results. ES and MAG prepared the manuscript with contributions from all co-authors.

**Competing interests**

The authors declare that they have no conflict of interest.

**Acknowledgments**

This work was carried out and data used in this manuscript was produced as part of the international Multidisciplinary drifting Observatory for the Study of the Arctic Climate (MOSAiC) with the tag MOSAiC20192020. We thank all persons involved

in the expedition of the Research Vessel *Polarstern* (Alfred-Wegener-Institut Helmholtz-Zentrum für Polar- und Meeresforschung, 2017) during MOSAiC in 2019–2020 (Project_ID: AWI_PS122_00) as listed in Nixdorf et al. (2021). We would especially like to acknowledge Marcel Nicolaus and Donald Perovich for their effort to coordinate the sea ice physics work during MOSAiC. We are grateful to Marcel Nicolaus for his effort to coordinate ROV work during MOSAiC. We thank Julia Regnery for the assistance with ROV multibeam sonar measurements and ROV co-piloting. We are grateful for the

assistance of Niklas Neckel in processing MOSAiC Helicopter-borne RGB orthomosaics.

ES, BAL, KVH, and MAG were supported by the Research Council of Norway through project HAVOC (grant no 280292). ES was also supported by Research Council of Norway project INTERAAC (grant no 328957), BAL and MAG by Research Council of Norway project CAATEX (grant no 280531) and MAG and MM by the Norwegian Polar Institute. MAG and MM received funding from the European Union's Horizon 2020 research and innovation programme (grant agreement No

101003826) via project CRiceS (Climate Relevant interactions and feedbacks: the key role of sea ice and Snow in the polar and global climate system). ROV operations, IM and CK were jointly supported by UKRI Natural Environment Research Council (NERC) and the German Federal Ministry of Education and Research (BMBF) through the Diatom ARCTIC project (BMBF Grant 03F0810A). ROV operations were further supported by the Helmholtz Infrastructure Initiative Frontiers in Arctic marine Monitoring (FRAM). PA was supported through the Alfred-Wegner-Institutes internal project AWI_ROV and

BMBF through the Diatom ARCTIC project (BMBF Grant 03F0810A) and the IceScan project (BMBF Grant 03F0916A). The authors thank Stefan Kern and an anonymous reviewer for their constructive suggestions that helped to improve the manuscript.

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
