# Peer review of "Observations of preferential summer melt of Arctic sea-ice ridge keels from repeated multibeam sonar surveys"

_The Cryosphere, 2023_

## Author Response (AR1)

**Author's response**
**Response to Reviewer 1 (Stefan Kern)**
*We thank the reviewer for their helpful comments and the work they put into the review. Our answers start with "**Reply:**" in bold following the original reviewers' comments.*

Summary:
This manuscript highlights investigations carried out in the context of the MOSAiC drift expedition of the R/V Polarstern, focussing on sea ice melt during summer (June/July) 2020. The novelty of this manuscript is the presentation, analysis and discussion of ROV-based upward looking multibeam sonar measurements under an assembly of first-year ice, second-year ice and ridges of different maturity and age. The aim to shed more light on sea ice bottom melt, particularly the difference between level ice and the keels of the ridges. Several factors that could investigate bottom melt related to the keel properties are taken into account and discussed.

To my opinion, this is an interesting piece of work that should be published. Before that I ask the authors to take into account a few issues I list below. I believe these would improve the readability of the manuscript, complement it in its content and provide an enhanced idea of next steps.

**Reply:** *Dear Stefan, we thank you for your positive and constructive evaluation of our manuscript. We have done our best to adapt your main concerns, including restructuring our introduction, providing more detailed description of our measurement techniques and instruments, and restructuring our "Results and discussion" section to make a clearer division between our results and their interpretation or intercomparison. We also updated our figures, adding a scheme of a ridge structure/morphology and some other minor corrections. We have provided detailed replies to all your questions and suggestions below, attempting to provide enough referenced evidence.*

General comments:
GC1: I find the introduction is written in a rather unorganized way and strongly recommend to structure it better. You will find some specific but also some editorial comments focusing on the introduction that might help you here. Also, you will find a number of questions that could assist you in adding more relevant information to the introduction so that it becomes a much better entry point to your manuscript.

**Reply:** *Thank you for your comments. We considered all your suggestions related to the restructuring of our introduction section and made the corresponding changes. It is currently structured as follows. First, we present a definition of an ice ridge (schematic is added to Fig. 1e), describing how widely they are spread within the Arctic, and why they are important. Second, we describe ice ridge classification and cover previous observations comparing melt rates of ridges and level ice. Next, we cover the development of sea ice observations performed using multibeam sonar, including level ice or ridges. Finally, we show that the current available observations of enhanced ridge melt are in fact quite limited, so we can substantially improve them using repeated multibeam sonar surveys.*

GC2: To my opinion, there are several aspects of the measurements themselves that deserved to be described in more detail - despite some of it might or even is published elsewhere. You will find these in the specific comments. The same applies to the introduction of the material you are working with: a ridge, how it is formed, how it looks like schematically and so forth. The terminology needs to be introduced better, some terminology seems to not be used in the right context and some terms and their definition and physical meaning for the work presented are even very unclear (e.g. keel front). Also for this part you will find several specific comments further below.

**Reply:** *Thank you for your suggestions. We considered most of them, including presenting more details about both sonar and coring observations. But we cannot fully understand what exactly we should add about ridge formation (its formation time is given in line 124, block size is in line 126, reference to temperature salinity profiles is in lines 148–150, the ridge photo is in Fig. 1c, the ridge areal image is in Fig. 2b). We feel that beyond this ridge formation processes are beyond the scope of this manuscript, and we do not have observations that cover that in any detail. We added a small description of salinity and snow mass fraction from isotope composition (lines 148–150), so a reader does not have to check this information*

*in the other publication. Similarly, we added methodology of our investigations of ridge consolidation during MOSAiC. For the terminology, related to the ridge parts (upstream and downstream flanks, keel bottom depth), they follow previous publications, particularly Skyllingstad et al. (2003) and Ekeberg et al. (2015) to be consistent and ease comparisons between studies. We also refined definition of keel front (lines 171–174).*

GC3: Section 3 currently is a melange of measurement results' descriptions, attempts to voice limitations and drawbacks, pieces of intercomparisons to previous work, and also some introduction of methodology used (i.e. how to convert draft changes observed into total ice melt estimates involving hydrostatic balance). Here I feel your manuscript has the greatest potential for improvement. Try to put everything that reads like part of the methodology into the respective section; try to be focussed on presenting the true results in section 3; try to come up with a new section 4 into which you put elements of the intercomparison and an extended version of the limitations and caveats of the measurements and analyses carried out. Your current subsection on the "limitations" part seems rather short to me.

**Reply:** *We thank you for your suggestion to improve the presentation, however, we chose a combined "Results and Discussion" section to streamline the writing and reading, and this is a matter of taste. We moved a few sentences from section 3 to section 2, mainly related to the methods and theory behind ice draft, thickness, and density values, as well as related to density measurements themselves. In section 3.6 we also divided our own measurements and their intercomparison with other observations. Yet, we are not convinced that dividing subsection 3.7–3.8 into separate sections improves the manuscript, as section 3 presents both results and discussions. We agree that it is helpful to present an outline of our study (lines 84–88) to discuss these issues one by one: in section 3.1 we present draft change of FYI, SYI, and a small ridge and provide a possible explanation why thinner ice experience smaller melt. In section 3.2 we also present draft change of the main ridge, as well as its main morphological characteristics. We also compare these characteristics with several previous studies to show which of them were representative for ridge observations. We also present draft changes of separate ridge parts. In section 3.3 we present results of linear regression of the ridge draft change and its cross-sectional characteristics, also mentioning potential reason of wider ridges having smaller melt. Section 3.4 provides estimates of surface and bottom accumulated melt for both level ice and ridges, while mentioning that differences between coring and sonar results will be discussed later. It also presents results of additional ridge parameters, which may influence its melt rates such as porosity. Section 3.5 is a very short overview of melt pond drainage and how they affect ice draft measurements. Here we refer to a few meltwater drainage studies, as its timing and volume are outside of the focus of our paper. Section 3.6 is dedicated to the effects of sea ice density on draft measurements. It also covers "true" measurements, both from coring and from sonar, but is using a different approach from section 3.4, where sea ice density was assumed constant as in many remote sensing studies. After revealing a more accurate estimate of ridge enhanced melt, we also provide estimates of ocean heat fluxes using widely used methodology. We conclude the "Results and Discussion" section, with section 3.7 discussing the intercomparison with previous observations, while section 3.8 describes study limitations.*

Specific comments:
L11: "40-50%" --> really? Where does this number come from?
**Reply:** *The sources are noted in the Introduction. This range of values are supported by upward-looking sonar measurements in the Beaufort Sea by Melling & Riedel (1996) and in the Fram Strait by Hansen et al. (2014), referenced in the introduction in lines 35–40, giving such range of ridge areal fraction by the end of the freezing season. The estimates from modeling usually give similar values but are likely less reliable (lines 33–35). Currently, the helicopter-borne or satellite altimetry cannot give an accurate estimate of ridge fraction partially due to complications of distinguishing ridge sails and snow accumulations (Duncan and Farrell, 2022, 10.1029/2022GL100272) and the fact that sails constitute a minor part of whole ridges.*

L15: "snow" --> To read about snow melt in the context of ice draft and underwater multibeam sonar surveys reads a bit strange. Also, I note that snow really play a very minor role in your entire manuscript. You are not really referring a lot to snow melt and/or how heterogeneous the snow cover potentially was.

*Reply: That is correct that snow plays a relatively minor role in our study due to its minor contribution to the hydrostatic balance of sea ice at the time of the advanced melt (June–July). We mention it here as snow depth was measured at ridge and FYI sites and was considered in our calculations with 0.08 m and 0.38 m of snow melt for FYI and ridge (lines 261–269), giving a substantial difference of draft change of around 0.13 m. Generally, MOSAiC observations from Itkin et al. (2022, 10.1525/elementa.2022.00048) show that ridges may accumulate 3 times more snow than level ice in spring, which also leads to a delay of ridge snow and sail melt, which is relevant for our observations. Despite snow not being the main contributor to the hydrostatic balance, we think that it is better to mention that 0.5 m of snow above the ridge was considered in our analysis.*

L18/19: "with the maximum point ice loss of 6 m" --> I was wondering how much of such loss isn't in fact caused by mechanical failure of the ice blocks building the keel and hence should perhaps not be listed under the term "melting".

*Reply: Thank you for this comment. We corrected the reported value of the maximum observed draft reduction from 6.1 m to 5.6 m as we excluded areas with possible mechanical erosion (lines 209–210). For the statement in line 18 we were not using areas with mechanical erosion. We also mentioned in the study (lines 248–249) that mechanical erosion was identified by uneven temporal evolution of ridge draft in the small area at y = 148–160 m with only 10% larger average draft decrease, than for areas without erosion. In our analysis of cross-sectional melt and point-measurement melt (Fig. 3b) we were not using any measurements from the area with possible mechanical erosion. For practical reasons in Fig. 3b we do not show outlined points, but there were 50-point measurements with the draft change within 4.0–5.6 m (equivalent to 4.4–6.2 m total melt), which allowed us to make the statement about the maximum point ice loss of approximately 6 m in the abstract. We also added a note that the ridge was fully consolidated which also supported the absence of mechanical erosion in the areas with trapezoidal keel shape (line 251).*

L19: "57% of the ridge total melt" --> does this refer to exclusively the under-water part of the ridge?

*Reply: No, that is why it is labelled as "ridge total melt" as it includes snow, surface (sail), and bottom (keel) ice melt, which are impossible to distinguish for each cross-section using only underwater sonar measurements of the ice draft. As suggested, we added that total melt includes both surface and bottom melt for clarity (line 19).*

L25-27: "Ridges consist ... in the rubble)" --> I recommend to add a schematic figure of how a ridge looks like - together with the definition of the quantities you refer later to in your manuscript such as width, slope, depth, and so forth. I invite you in the same context to clarify whether every ridge has a keel and/or a sail and also comment on the relationship between sail height and keel depth.

*Reply: We added a simple schematic of a sea-ice ridge structure to Fig. 1e. We also removed unused "rubble" and added a definition of a consolidated layer (lines 27–28). The detailed underwater characteristics of the ridge were given in Fig. 4a, including keel draft, slope, bottom width, and distance to keel front (right next to the results about their contribution to ridge melt). According to WMO definition, each ridge has a keel and sail (line 24). We try to avoid referring to previous reports of ridge to sail thickness ratio, as we believe they may confuse the reader. The commonly cited keel to sail ratio of 3–4 is often based on incorrect or confusing reporting, and usage of only maximum values. The relationship between maximum thickness of keel and sail is defined by the ridge hydrostatic balance, and since this balance depends on their cross-sectional areas, equals to the square root of 9 (ratio of level ice draft and freeboard) or around 3, with some corrections coming from the snow depth pushing ridges downwards. Ridge surveys including all values of keel and sail thicknesses for the same coverage (not just extremes), usually give the ratio of keel and sail as 9 to 1 like for level ice, as we show in Fig. A1 and line 309. Since maximum sail height plays a minor role in our analysis, we decided not to include information about keel and sail relationship in our introduction.*

L29: "during the melt season" --> How become water-filled voids sea ice during the melt season?

*Reply: The study from Marchenko mainly provides short-term observations of fully consolidated ridges near Svalbard during melt season, but the exact timing and process of such consolidation were not known.*

*In brief, fresher meltwater can refreeze because the temperature of the rubble in the keel is lower than the freezing point of meltwater. We now provide the potential explanation of this process in lines 29–33, with reference to our earlier work from MOSAiC.*

L29-33: Here you go on describing areal and volume properties - based on two studies. I was wondering whether there is not more information available. Is the Rothrock work based on observations or is this model data as well? I was wondering whether you could give more explicit examples about the importance of ridges for climate studies or other fields of concern, such as e.g. shipping. What makes ridges interesting? How much research has been devoted especially to the importance of keels? What specific roles do keels play? How much of the additional ice volume contributed by sea ice ridges is actually located in the keels?
**Reply:** *We added a clarification that Rothrock estimates are also based on "ice-ocean coupled modelling". There are also other estimates of ridge areal fraction presented based on upward-looking sonar measurements (lines 38–45). We also moved information about the importance of ridges for biology and underwater light conditions. The fraction of "additional" ice volume in ridges can be directly calculated from Mårtensson et al. (2012) as 27%. We opt to keep the focus on topics that are most relevant to this study. This is not supposed to be a review of the sea ice ridges, and we believe we have given more than enough pertinent information in the Introduction that is relevant for this study and its motivation. And it is obvious that studies focused on the sea ice ridges are very few, compared to those that focus on level ice.*

I don't get why you need to start a new paragraph in L33.
**Reply:** *We agree and removed this text "break" alongside all other major edits made to the Introduction.*

L48-50: "single-point ... early October 1998." --> Would you be able to elaborate on this kind of observations with just one more sentence explaining how this was realized? Were these ablation stakes penetrating the entire ice column, i.e. reaching well into the water, and were read then from below the ice with a camera or by a diver? Or was this an indirect method where the ablation stakes were only put into the ice and any bottom ablation amount was inferred from the change in sea ice freeboard in combination with similarly observed snow depth? I find this rather important in light of the various attempts to compare your findings with these earlier findings throughout this manuscript - also when it comes to OHF estimates.
**Reply:** *We initially used a short name of observations developed by Perovich et al. as "stakes", but now we added an extended name as "hot-wire thickness gauges", which directly measure bottom ice growth or melt. It is already mentioned in this sentence that Perovich et al. "measured bottom melt" using these devices, meaning that it is not an estimate but a direct measurement of ice loss at the bottom using a hot-wire. In brief, a hot-wire thickness gauge is a bar connected to a steel wire which can be heated by a generator so the wire can be pulled upwards which allows to measure bottom growth/melt. We believe that measurements of ice bottom melt from ice drilling, gauges, or IMBs are nearly identical, and their main limitation is related to the spatial variability of sea ice processes, which we attempt to show in this manuscript.*

L99/100: I am not an expert in the vertical extent of different water masses in the Fram Strait but when I look at panel c) of Fig. 1 then I would state that the CO entered deeper water (again) after July 14 after having been in comparably shallow waters during the period June 24 to July 14. Hence I am not convinced that while drifting South through the Fram Strait the CO had a shallower Atlantic water layer underneath. You might want to check your writing.
**Reply:** *We are sorry for the misunderstanding, but we were not saying that MOSAiC floe had Atlantic waters directly underneath but quite opposite. We note that the Atlantic water was at a shallower depth, but the surface layer was still of cold Arctic character throughout the observation period and ocean heat fluxes were low and we did not have an excess heat source from Atlantic water (making the observations still comparable to the conditions in the central basin). We added a reference to the overview of oceanographic work during MOSAiC from Schulz et al. (2023) which describes this in more detail (lines 106–109). Details beyond this do not really serve this manuscript.*

L103-106: The ridge names used in this paragraph should be introduced before.

**Reply:** *We do not quite see why these names should be given earlier without proper context, except here in the first paragraph of materials and methods. We further added in line 115, that this study is focused on Jaridge, one of the two mentioned ridges, while investigations of Alli's Ridge are covered in a different manuscript with a different focus. The main reason for this MOSAiC ridge overview was to show that there are no conflicting/overlapping observations of the ridge melt.*

What I also don't understand: Wasn't there a break in the observations in June with the Polarstern exiting the ice in May/June and then getting back into the ice in June again to eventually continue with the work. Was it therefore really possible to study that ridge named "Alli's ridge" all time from January onwards? Please clarify in the text if possible.

**Reply:** *We added a clarification that the Alli's Ridge was studied "during January–February and June–July" (lines 112–113). Yet it was indeed surveyed continuously using two Ice Mass Balance (temperature) buoys during January–July, but there were no coring/drilling measurements during the break in May. However, this period is outside of the advanced melting season with the multibeam surveys, it is not a limitation in this study, and this is given to provide complete context of the MOSAiC campaign.*

Fig. 1 a) What explains the decrease in FYI bottom temperature between July 20 and July 30?

**Reply:** *This decrease in FYI bottom temperature was caused by stronger mixing of the under-ice meltwater layer with colder seawater as false bottoms "protecting" this meltwater stratification started to melt (due to the increase of ocean heat fluxes). We presented a more detailed study on under-ice meltwater evolution (Salganik et al., 2023, 10.1525/elementa.2022.00035), while the presence of such layer is mentioned here in lines 285–286.*

L119/120: What was the motivation to use a) an inconsistent number of drillings per transect and to use b) an inconsistent mixture of horizontal spacings of drillings and other measurements along the transects. Also: Why were the different transects placed at the locations displayed in Fig. 2b)? Why didn't you chose a more regular pattern and/or even used more transects?

**Reply:** *It was not a motivation/plan, but a necessity due to the conditions and challenges encountered in the field. In brief, we usually started at the sail crest and proceed sideways. Ones we reached level ice, we started a new drilling line. Lines 1, 2, 3, and 5 have the same distance between each other of 5 m. Line 4 was added in between lines 3 and 5 as we approached the end of the ridge area with a pronounced sail and trapezoidal keel shape. Lines 6 and 7 are located further where the ridge sail started to be pronounced again after skipping a small region at y = 150–160 m (later identified as the region with disconnected blocks prone to mechanical erosion). Lines 1–3 were revisited as the lines with non-zero initial macroporosities. Other lines were not revisited mainly due to logistical limitations (the ice floe broke apart in late July). The initially planned distance between adjacent drill holes of 5 m was found to be too large for this relatively narrow ridge, so we decided to reduce it to 2.5 m to avoid double-work. However, the drillings mainly provide a set of calibration/validation data points for the sonar surveys, for that its more important we have drilling within the sonar surveys, than where exactly they are located.*

L120: "The ridge was measured seven times ..." --> Does this mean that every transect shown in Fig. 2b) was measured on a different date? If so, how representative are these measurements at different places then for the overall structure of the ridge? Is the ridge that homogeneous?

**Reply:** *The exact dates of our drilling measurements are mentioned in figures showing drilling results (Fig. 2a, A1a-f). Usually, 2–3 adjacent lines were drilled during each of the mentioned 7 days. We added a total number of the performed drillings in line 131 ("with a total of 47 drill holes"), already mentioned in line 273. The aims of drilling measurements were (1) to validate sonar draft measurements (which in turn covered the whole ridge on every survey, irrespective whether drillings did not cover all transects completely enough data points are available); (2) to provide an estimate of ridge macroporosity; and (3) to collect measurements of sea ice salinity, temperature, and density for an accurate thermodynamic/hydrostatic analysis. Drilling lines 1–3 were revisited, other drilling lines initially contained almost zero voids (so porosity reached its minimum). The homogeneity of ridge internal structure is a complex topic (and beyond the scope of this study), but we showed that the sampled drill holes were*

*strongly consolidated with an average macroporosity of only 4±7 % (line 249) substantially lower than typical values of 30% for first-year ice ridges in winter (line 28). Having these measurements increases the accuracy of the estimate of melted ice volume. Additionally, given the ridge was consolidated this reduces the probability of mechanical erosion as a reason for the observed draft changes.*

Figure 2:
- Panel a) denotes points P2 and P3. What are these standing for?
**Reply:** *In section 2.3 we explained our assumed (trapezoidal) shape of ridge cross-section, with $P_1$–$P_4$ being four corners of each trapezoid with upstream flank $P_1$–$P_2$ and downstream flank $P_3$–$P_4$. We added an explanation of this concept to the Fig. 2 caption.*

- If the gaps in the blue vertical lines denoting the drilling on July 3 are meant to denote voids in the ridge I suggest to mention this in the caption. I also recommend to describe the drilling process in more detail in the text. How did you technically figure out that there were voids along the cored section and how accurate is their vertical location? How did you actually measure the total thickness and with which accuracy? Did you take out the ice cores (possibly not, these are too small in diameter) so you possibly measured with a ruler tape. How?
**Reply:** *We added to Fig. 2 and A1 a caption that "The vertical lines are drill holes, the solid lines are ice, and the line breaks are voids". We also added in line 128 that we used "thickness tape to measure snow or ice interface position". Ice drilling with 2-inch augers is a standard way to measure ice draft and freeboard, including ridge observations (Strub-Klein and Sudom, 2012, 10.1016/j.coldregions.2012.05.014). And it is quite common not to describe this process in nitty-gritty detail even in studies entirely focused on ridge drilling (Leppäranta et al., 1995, 10.1016/0165-232X(94)00019-T). The position of ice-water interfaces is registered using a thickness tape placed parallel to the drill. The precision is usually 1 cm. The total ice thickness is additionally measured with a thickness gauge with a similar precision, but arguably smaller errors. The exact values of such errors are usually not mentioned in papers probably due to errors mostly defined by the environmental conditions (weather, presence of slush, ridge size) and experience of operators. We extracted several 7–9 cm diameter ice cores from 5 places of this ridge, with salinity, temperature, and isotope compositions presented in Lange et al. (2023, line 149), with coring locations indicated in Fig. A2. These measurements are not relevant for our estimates, so we do not discuss them in detail. The density measurements used in our study are presented in this study in the form of gas volume fraction (Fig. 5b) and bulk sea-ice density (Fig. 5d).*

- Check usage of terms "upstream and downstream". To me upstream is looking "upwards", e.g. to the direction where the stream (or drift) comes from while downstream is looking "downwards", i.e. into the direction the stream flows. Usage of these terms you might need to revise also in later figures / text related to them.
**Reply:** *That is precisely how flanks are defined in our study with water first reaching the "upstream" flank and afterwards the "downstream" flank after passing the keel (Fig. 2a), following concept of river upstream and downstream parts and labelling from Skyllingstad et al. (2003). The ice drift direction is opposite to the direction of the water motion relative to the ice, and what matters in this study is the water motion relative to the keel.*

- It is not sufficiently clear what the ragged white vertical lines in panel b) mean. Are these vertical profiles?
**Reply:** *We added to the figure caption that "keel width boundaries" are labeled as "$P_1$ and $P_4$". These vertical lines are accompanied by $P_1$ and $P_4$ labels, which are also shown in Fig. 2a as keel width boundaries and mentioned in the text in line 168.*

- I recommend to switch colors in panel c). The really interesting part is the draft and this is the parameter that should be colored here. The outlines of the different ice types can be realized by white, grey and black lines, using dotted or dashed linestyles in addition.
**Reply:** *This color scheme was chosen intentionally to show the difference between the main ice types in other figures using the same color scheme and to have a good readability of ice classification. We provide a*

*more detailed draft change plot (Fig. 4c) and blue-white-red contour plot of the ridge draft (Fig. A2). The colors of ice type areas were chosen to match the colors of Fig. 3abc and Fig 1a. Please indicate if you think that Fig. A2 should be moved to the main part of the manuscript.*

- If possible make panel b) to have the same aspect ratio as the dotted box in panel c) denoting this subregion.
**Reply:** *We thank you for your suggestion and adjusted the coverage of Fig. 3c, so the white-dotted-line frame fits the aspect ratio of Fig. 3b.*

- Please note what type of aerial image is used in panel b).
**Reply:** *We added that it was "optical helicopter-borne" aerial image.*

L142-144: "We use measurements ... of 0.5 m." --> I note that a more detailed description of the instrument is given in Katlein et al. (2017). However, how the surveys with this device were carried out in this particular case is not sufficiently well described. The seven surveys you write of seem to be distributed in time and not in space. Hence, how was every single survey designed? Was a mattress like pattern scanned? If yes, what was the distance of the adjacent scans. How wide is the imaged area during one multibeam sonar scan from a depth of 20 m? Do adjacent scans overlap? How long does each of these seven surveys take and how were they evaluated, i.e. how did you end up with the reported 0.05 m draft accuracy?
**Reply:** *We added more details about the sonar surveys to lines 154–156 saying that "The surveys were done in a grid pattern with distance between lines of 20–25 m. The sonar has 480 beams with an across track swath width of 120° (64 m width for level ice) and an along swath width of 3°, an effective beam width of 0.75°, and the angular resolution of 0.25°". Coppolaro (2018, doi:10.13140/RG.2.2.34572.95362) estimated spatial resolution of the same multibeam sonar as 0.03–0.04 m. The DT101 sonar heave accuracy is limited by 5 cm (imagenex.com/products/dt101xi). The survey duration was around 2–3 hours.*

L152: What do you mean by "keel front"?
**Reply:** *The keel front is a straight-line tangent to a polygon of all $P_2$ locations (lines 171–172, Fig. 4c). For each ridge cross-section, $P_2$ is the bottom corner of the ridge upstream flank.*

L155/156: "The upstream ... relative to the ice" --> Please see my comment to these terms in Fig. 1. I guess you are perhaps mixing two issues here: the wind-driven movement of the ice (i.e. a force above the sea ice) and the prevailing ocean current (i.e. a force underneath the sea ice). I suggest to use the terms "upstream" and "downstream" only in relation to the dominant forcing and hence drift - which possibly is the wind forcing.
**Reply:** *In our study the dominant forcing providing energy for ice melting comes from the ocean and we label our flanks relative to the flow of the water relative to the ridge. The hypothesis was to compare melt rates of upstream and downstream flanks, where downstream flank may experience more turbulence and therefore more ice melt. This concept was initially suggested by Skyllingstad et al. (2003) using the large-eddy simulation. They used similar labelling as "Perturbations in the flow downstream from the jump actively mix the stratified layer, as shown by visually comparing the vertical gradient of salinity in the lee of the keel with the upstream conditions". Therefore, we prefer to use similar labels of ridge flanks.*

L160: I don't understand this sentence. If the keel front is defined as the tangential line or plane at P2 then the distance from P2 of each cross section to the keel front should ideally be zero ...
**Reply:** *The keel front is a straight line, while corners of each of 131 ridge cross-sections form a polygon, shown in Fig. 4c and labeled as $P_2$. The keel front is also shown in Fig. 4c as red dashed line. A distance between $P_2$ polygon and keel front line is one of the factors we used for ridge melt analysis. The distance to keel front is also shown in Fig 4a.*

And how can "the distance from P2" used as one of the factors to estimate ridge melt? This needs to be explained in more detail.
**Reply:** *We answered this question above.*

L200: I still have difficulties to understand what you mean by "distance from the keel front" and what the physical mechanism is behind this "parameter" having an impact on ridge melt variability.

*Reply: The proposed process is described in lines 246–247 as areas with larger distance from the keel front "were protected by the keel front from the turbulent fluxes, which appear to occur in the vicinity of ridge bottom corners ($P_2$ and $P_3$ in Fig.* **Error! Reference source not found.***a)".*

Figure 4: You write that the red dashed line denotes the keel front. I still don't understand how this is defined and at which time this was defined as being the keel front. Apparently, it was defined for a date earlier than June 24 - otherwise it would not make sense why the ridge front is located to the left of the blue draft profile in Fig. 4 a.

*Reply: This is not correct; it was defined for the first sonar scan from June 24 (added to line 172). The keel front shown here as red dashed line labelled as "keel front" is a tangent straight line, touching the ridge bottom corner of upstream flank shown as black line and labelled as $P_2$. Blue shaded areas highlight ridge cross-section with a large distance between keel front and $P_2$. Some ridge cross-sections apparently may have zero distance to the keel front, while on average it is positive as shown in Fig. 4a. Unfortunately, we do not understand how to make this description clearer.*

L226/227: "the sea-ice draft ... by water density." --> Please include an equation which describes this relation - so that it becomes more clear why you seem to relate a distance to something that seems to be related to a volume. Also "amount" is somewhat unspecific.

*Reply: We added a reference to Fons et al. (2023) with this exact equation. We feel that an equation illustrating a balance of gravity and buoyancy forces is unnecessary here for several reasons. First, it would require adding variables for each term, which are not going to be used thereafter. Second, we devote the following section to explaining why such equation can give errors when sea-ice density is time-dependent. We substituted "amount" by "thickness" to make the sentence more specific. If you think that the equation is still necessary, we may add it as following $\Delta h_{si}\rho_{si} + \Delta h_{sn}\rho_{sn} = \Delta d_{si}\rho_{sw}$.*

L230/231: "hence provided ... (0.25m)." --> I suggest to explain to the reader where this estimate of 0.25m comes from. It would be important also to note whether the surface melt was in this case 0.18 m as well. Finally, as written it seems the sea ice was free of snow. I can see that this is taken up in the next paragraph but it would be good to know whether the 0.18m mentioned in L229 was pure sea ice surface melt or whether this included snow melt as well. This is unclear.

*Reply: Thank you for this comment. We added here a clarification, that we assume the same snow and ice surface melt for sonar measurements as for FYI coring site. In the previous sentence (lines 261–263) we mentioned that the draft change from FYI coring was 0.34 m, which is (0.07 m) larger than 0.41 m mentioned in this sentence (line 264). We also added the amount of melted snow to line 262.*

L232/233: How can temperature measurements from IMB indicate a surface ridge melt of a certain quantity? As far as I recall such measurements the temperature profile near the ice/snow interface becomes rather blurry and it is not that straightforward to define precisely where this interface is located. I invite you to provide more details in this respect.

*Reply: Thank you for this question. That is correct that only in situ temperature measurements from an ice mass balance buoy would not allow for a high accuracy of snow-ice interface detection in summer. Here we also used "heating-induced temperature difference" measurements (added to line 266) which allow to distinguish materials with different effective heat capacities with accuracy of sensors vertical resolution of 2 cm. We added this more detailed description with a reference to lines 145–147 ("... and daily heating induced temperature difference measurements after a cycle of internal heating allowing to identify the location of snow-ice and ice-water interfaces with high precision (Jackson et al., 2013)".*

L233-235: "Assuming 0.24 m ... 0.93 m" --> Please provide together with the equation I asked for earlier also the values used for the densities to be able to convert the measured changes in vertical dimensions into volume and then back in a vertical distance. Thank you.

**Reply:** *Thank you for your comment. We added a value of snow density above MOSAiC ridges in June–July together with a reference of the dataset, where it was reported (line 268). We also made a reference for the equation (see above), but here all the values represent thicknesses, not draft changes, so it is not needed here.*

L274-279: "Measurements from an ... Arctic summer" --> I have five questions here.

1) How accurate are the measurements of the airborne laser scanner of which one was carried out July 4 and the second one July 17?
**Reply:** *We added that the accuracy of the used Riegl VQ-580 airborne laser scanner is 0.025 m (Ricker et al., doi.org/10.5194/egusphere-2022-1122) which is enough to identify the 11 cm imbalance between level ice melt and draft change, presented in our study (line 306, Fig. 5c).*

2) Further up you wrote about the melt-pond drainage induced increase in freeboard by 8 cm from July 9-13 followed by a rebound of x cm from July 13-17. This falls well in the period you are looking at here.
**Reply:** *We do not completely understand this remark. That is correct that we observed a strong and rapid freeboard increase during 7–14 July right after the melt pond drainage on 9–13 July, while thereafter the melt pond fraction was recovering during 13–17 July. Therefore, laser scanner measurements on 4 July and on 17 July were not able to capture this short-lived uplift due to melt pond drainage, but only a long-term uplift due to FYI density decrease. We note it since it occurred in the period of interest.*

3) I am wondering about the physical mechanism that is able to cause the observed / hypothesized FYI density change. Brine pockets merge and brain drains downward and is also flushed by the surface meltwater. I would expect that most of the FYI above the water line was saturated with melt water before the drainage event became porous shortly after it, but with continued melt the majority of the voids should then again quickly be filled with melt water. In short, I am wondering where this large air-fraction comes from.
**Reply:** *The recent work based by Crabeck et al. (2016, 10.5194/tc-10-1125-2016) based on computed tomography showed that processes of salt redistribution do not control the distribution of gas in sea ice. And, indeed, most of the gas volume was located below the waterline. Yet, there is little known about the exact mechanism of how gas fluxes or biological activity affect sea ice gas fraction, and this is beyond the scope of our study to speculate on this further, what is relevant is what the density data tells us. For us, the most important is that our density estimates agree with both historical data of sea-ice density and with measured draft to thickness ratios.*

4) How was - during the coring and drilling process - ensured that only a small (ideally constant) amount of any liquid (be it brine or melt water) drain out of the core sections before they were put into boxes or bags for melt? Could your FYI density measurements in any kind be biased by changes and/or variations in the attention that was given to this process?
**Reply:** *The density cores were the last cores to be collected during the coring program. They were put together with other relatively cold cores in an insulated box without sectioning to minimize brine losses and immediately transferred to the cold lab located in less than 1 km. The ice in-situ temperatures were similar during our observation period in June–July. The sea-ice density was measured at –15°C and, generally, depends mostly on gas volume, not brine volume / salinity. Most of the gas bubbles were located at a specific area slightly below the waterline (with the lowest ice temperature and permeability). These gas bubbles were symmetrical, densely packed and were very visible right after core extraction, they were different from brine channels or pockets. Meanwhile, the reported density values are very similar to historical observations compiled by Fons et al. (2023). Additionally, a slight increase of sea ice density right after melt pond drainage recorded on 13 July agrees with observations of meltwater blockage by Polashenski et al. (2017, 10.1002/2016JC011994). A similar coring program was performed at a second-year ice site and at several ridges with identical handling but without similar gas fraction increase. The MOSAiC coring program is described in more detail by Salganik et al. (2022, 10.1525/elementa.2022.00035). We are planning to validate our coring measurements using computer*

*microtomography. Currently, there is no obvious method to assess the potential errors of such measurements. That is why we compare them with freeboard measurements from ice coring (Fig. 5d) and airborne laser scanner (line 300). Pustogvar and Kulyakhtin (2016) estimated an error of laboratory hydrostatic measurements of sea-ice density below 2 % (line 192), which is more than 2 times lower than density decrease in out study.*

5) How much can you exclude that the freeboard increase observed by the laser and at the coring site isn't the result of the whole floe-system studied being in hydrostatic equilibrium with the level parts being by no means independent from the ridges and hence being kind of "carried" by these?

**Reply:** *We measured freeboard increase using laser scanner for a large area of first-year level ice (200 m by 100 m), similarly for the first-year level ice coring site, but for smaller area. We are not sure which other types of ice may carry this dominating (first-year level ice) type of ice (Fig. 2c). We did not observe freeboard increase for ridges; we also did not measure a substantial decrease of the ridge density. And we fully agree that sea ice is flexible and can easily allow for vertical deformations of around 10 cm on the horizontal scale of several tens of meters. Therefore, each ice type (FYI and ridge) is assumed to be in hydrostatic equilibrium at the sufficiently large scales that we are looking at here.*

L302/303: "Despite ... only 60%" --> please again back this up with the respective reference. In general, for this sentence, it might make sense to state that these were both experiments carried out at one isolated ice floe and therefore are of case-study character. It might not be justified sufficiently well to say that this reflects the general conditions.

**Reply:** *Thank you for your comment, we added a reference to the study by Perovich et al. (2003), line 340. We showed in the introduction of this study, that any high-resolution observation of under-ice topography evolution is limited by relatively small scales. Unfortunately, other methods cannot provide similar estimates with comparable errors. We dedicated section 3.8 to covering limitations of our and earlier studies.*

L307-311: Here and also in other places of your manuscript, you are writing quite a bit about vertical OHF. I was wondering how well one could intercompare the numbers used here because, as I see it, there is a great variety of how the various authors - including you - ended up at the numbers presented. It appears to me that during the old SHEBA experiment OHF was actually measured while in all the other cases mentioned by you the OHF was estimated - possibly involving a number of assumptions and also involving different types of measurements to track the progress of bottom melt. In short: How comparable are all these values?

**Reply:** *It is not correct that ocean heat fluxes during melt season of SHEBA were estimated differently from our observations. The values we present following Perovich et al. (2003) are based on hot-wire gauges measurements of actual ice bottom melt. In our study we similarly use ice bottom melt measurements from coring and buoys as well as estimates from sonar (assuming the same surface melt as for coring). The same commonly accepted approach was used by Shestov et al. (2018) for N-ICE 2015 ridge studies based on thermistor string data. Krishfield (2005) used an indirect approach (parametrization) using only seawater salinity and temperature measurements to upscale estimates of OHF for the whole Arctic Ocean. Of course, OHF may differ depending on the vertical scale and range, and for that reason we also refer to the direct oceanographic measurements of ocean heat flux (turbulence measurements) performed during MOSAiC (Schulz et al., 2023, 10.31223/X5TT2W). Therefore, the values in our study are intercomparable and represent estimates of how much energy was spent on actual sea ice melt, with accuracy mostly limited by accuracy of ice thickness measurements, since during melt season other heat fluxes (apart from latent and ocean) at the ice-water interface are nearly zero.*

Also, since you are mentioning proximity to Atlantic Water in case of the N-ICE2015 campaign I was wondering whether a figure that compares locations of bottom melt observations and observed depth and temperature of the Atlantic Water potentially involved could shed more light on the different impact of oceanic conditions.

**Reply:** *We feel this is beyond the scope of our study (such an estimate is be presented for MOSAiC drift by Shultz et al., with very low ocean heat fluxes (2.1 W m⁻²) below the mixed layer, lines 108–109). For the upscaling, we may refer to the study by Krishfield (2005) providing estimates of OHF for Arctic Ocean. In our case, the OHF was related to seasonal changes in solar radiation and ice concentration, with OHF values similar to both SHEBA and Central Arctic following Krishfield (2005), which was not the case for N-ICE2015 (Peterson et al., 2017), where they actually provide such map with OHF variability (as it is relevant for that region). Therefore, we suggest that our observations are representative for Central Arctic conditions with low heat fluxes over the halocline.*

**Typos / editorial comments:**
L15/16: We investigated "an" Arctic first-year ice ridge ... bottom melt rates of "the" ridge keel ...
**Reply:** *Modified accordingly.*

L37: "using draft measurements from" --> perhaps better in the context of this sentence: "derived from draft measurements by"
**Reply:** *Modified accordingly.*

L39: "...1990 to 2020. Sea-ice ridges ..." --> Here might actually be a good location to split the paragraph because above you refer in general to ridge fractions citing various sources while below you get back to actually describing how ridges are composed and/or formed.
**Reply:** *Thank you. It is modified accordingly.*

L41: "as" --> "at"
**Reply:** *Here we tried to say, that thin ice is the weakest point itself (and therefore ridges are often formed from thin ice), not that it is breaking at the weakest point of thin ice.*

L47: "rather deep ridge" --> Just to clarify: Are you referring to the vertical dimension of the entire ridge, i.e. sail + keel, or just the keel draft here? I was puzzled by usage of the term "deep".
**Reply:** *We agree with your comment and substituted "rather deep ridge" to "large ridge with the maximum total thickness of 10–12 m", as these values refer to the sum of sail and keel thicknesses as reported by the authors (line 53).*

L51: "winter season ... expedition" --> perhaps better "during the SHEBA expedition in winter"
**Reply:** *Here we tried to underline, that SHEBA expedition covered all seasons, but these specific measurements were performed in winter.*

Furthermore: I am not sure how the winter measurements mentioned in the first part of the sentence refer to the statement that follows, focusing on summer conditions.
**Reply:** *We think that these are unique turbulent measurements, which were not realized even during MOSAiC due to logistical issues. Despite no substantial ice melt during those measurements, the turbulence pattern relative to ridges might be the same in summer (as also suggested by Skyllingstad et al., 2003).*

Another thing I need to ask: Were these really measurements or again just estimates based on bottom melt rates?
**Reply:** *We added clarification that these values from Skyllingstad et al. (2003) are made "using high-frequency measurements of seawater temperature, salinity, and velocity measurements" (lines 56–68). We also do not fully understand why generally precise measurements of ice bottom melt, providing estimates of the energy balance at the ice-water interface are not considered "real measurements" in comparison to parametrizations based on measurements of water velocity, temperature, and salinity that are more indirect measures of ice melt. For example, by McPhee et al. (doi: 10.1029/2007JC004383) directly intercompare both types of ocean heat flux measurements.*

L53-54: "from ice-profiling sonars mounted on subsea moorings in the Beaufort Sea" --> Is this by chance the BGEP moored upward looking sonar array? Then I suggest to write: "from BGEP moored upward looking sonar"; if not you can omit the BGEP but then name these instruments similar to those mentioned above in the context of the Hansen et al 2014 paper.

*Reply: These two instruments were located substantially closer to the land in comparison to the four BGEP installations. Moreover, the paper from Hansen et al. (2014, 10.1029/98JC01257) cited in our study covers Fram Strait sonar observations, not the ones in the Beaufort Sea. Therefore, we kept the original version of this sentence.*

L54: "their data does not ..." --> Why? Consider adding the reason.

*Reply: We mentioned in the same sentence that these estimates are based on upward sonar mooring measurements, which implies that it cannot measure the same ice twice (in their manuscript it is said that "With onshore and offshore motion superimposed on a general westward drift, the ice that moves back over the moorings was formed further to the east than the original, but nonetheless from the same band as that initially viewed"). We added "due to sideway ice drift" to this sentence to mention the reason of potential errors (line 62).*

L60: "Thus" --> "In summary" ?
*Reply: Modified accordingly.*

L61/62: "... a ... investigations" --> a ... investigation"
*Reply: Modified accordingly.*

L70-73: These lines should be moved to the location where you describe the importance of ridges in the (Arctic) climate system. Having followed my previous comments you will know where this is.
*Reply: Modified accordingly.*

L73-L77: "The measurements collected ... Lange et al. 2023)" --> This parts seems to belong to the sentence in L29 where you write about consolidation of ridges during summer melt.
*Reply: Modified accordingly.*

L87: I might not harm to, at this point, provide a brief outline of the structure of the manuscript.
*Reply: Thank you for the great suggestion. We added a sentence covering all seven sections of our results and discussions (lines 84–88).*

L134: "chain" --> "thermistor chain"
*Reply: Modified accordingly.*

L176/177: "The maximum ... of 6.1m." --> Please check the second part of this sentence; I guess something is missing.
*Reply: Thank you, we substituted "of" to "was" in this sentence.*

L182/183: "According to the ... in comparison to ..." Please check grammar of this sentence, it reads incomplete.
*Reply: Thank you, we added "melt of" before "keel bottom" in this sentence to make a reference to "the melt of ridge flanks" clearer.*

L241-243: "A study focused ..." --> While interested readers and readers with sufficient time will certainly prefer to read the paper cited here, there are others - including myself - who would appreciate to see one sentence that explains how the degree of consolidation of the ridge was measured and quantified (drilling? tomography? indirect methods?).
*Reply: We agree with your suggestions and added that that study is "based on observations from IMBs (ice mass balance buoys) and ice drilling".*

L245: I doubt the "abnormal" is needed here. The processes involved during the melting season have positive freeboard changes as one of their normal result.

**Reply:** *We cannot agree that the freeboard increase during melting season is a "normal" and well-documented process, which is included in sea-ice models or retrievals. For example, freeboard observations from satellite altimetry allows to track sea-ice decay as shown by Landy et al. (2022, 10.1038/s41586-022-05058-5), with the main errors currently coming from radar penetration through the warm snow, not due to meltwater drainage or sea-ice density evolution. That is why we mentioned that our observations can be useful for altimetry and remote measurements from upward-looking sonars (line 324).*

L264/265: When did this snow melt happen? Before June 20?

**Reply:** *We clarified that "the snow at the FYI coring site melted entirely by 20 July from an initial depth of 0.08 m" (line 291). The exact date of when snow depth became zero is quite uncertain as it requires knowledge of whether it was a precipitated snow or surface scattering layer (Smith et al., 10.1029/2022GL098349).*

L284/285: "...modification of draft to thickness ratio for analysis ... " --> I don't understand what you want to say here ... Consider rephrasing.

**Reply:** *We agree and rephrased this sentence as "These measurements suggest that considering draft to thickness ratio dependence on sea-ice density evolution of different ice types may improve estimates of ice melt from sonar surveys", lines 313–314.*

L287: "that" --> "than"

**Reply:** *Modified accordingly.*

L292: You might want to back up the "for lower sea-ice concentrations" by refering to some respective information of that parameter, please.

**Reply:** *We added a reference and value of sea-ice concentration estimates around MOSAiC of 85% for late July from Krumpen et al. (2021).*

L315: "are based on an assumption ..." --> Was this assumption not backed up by the ULS draft measurements? I mean, by tracking the draft over time and relating it with the actually present drift one can get a proper idea about whether draft has changed and to what degree.

**Reply:** *Changes of ice draft distribution during melt season include both spatial differences of ice draft and seasonal changes related to ice melt (and dynamics). Therefore, for that analysis the ridge fraction was assumed constant which may not be the case, as ice was drifting not only parallel to the axis (perpendicular to the shoreline), but also sideways, and the dimensions of the ridge keels can be distorted when one cannot know the ridge crossed the sonar perpendicularly, and the observations did not repeatedly measure the same ice, as we did here. In retrospect these shortcomings can result in rather large uncertainties, than for us, where we surveyed with certainty the same patch of ice.*

L318: If you would write the total level ice melt rate for MOSAiC with the same number of digits than you write the rate for Amundrud et al then you end up with 0.02 ... I strongly suggest to term that these are similar - within the error margins.

**Reply:** *We agree with your comment and substituted "higher than" to "similar to".*

L323: "0.9 m" over which time period?

**Reply:** *Thank you for this question, we added here that the draft decrease was observed by 26 July.*

L332/333: I am a bit puzzled that a description of how second- and multiyear ice ridges looked during SHEBA are cited with a reference (Kovacs 1973) dating before the expedition actually took place.

**Reply:** *We agree with your comment. Unfortunately, the description of ridges from SHEBA are quite limited (Perovich et al., 2003), so we referred to detailed observations of different second-year ice ridges from Kovacs et al. (1973). To avoid such confusion, we clarified the difference in lines 342–344.*

L349: "on average" refers to both the experimental site as a whole and the time period of one month? Please clarify.

**Reply:** *We added a clarification that "an average draft" mentioned in line 392 was "initial". We are not sure how the accumulated ice melt can be averaged over time, as we specify our measurements in line 389 as "change of sea-ice draft over a period of one month".*

L352: "Ridge bottom ..." --> This sentence requires additional explanation because a 4 times higher bottom melt for ridges and a similar surface melt does not correspond well to a factor of two the average total ridge melt is larger than the average total level first-year ice melt.

**Reply:** *Thank you for your comment. We added the range of 3–4 to line 393. In line 404 we mention that considering sea-ice density we refined "the ratio of total ridge to first-year level ice melt to 3.0". We added there that we also refined the ratio of bottom ice melt to 3.8 using density measurements.*

L355: I note that further up in your manuscript contributions from keel width and distance from keel front had a negative sign.

**Reply:** *Thanks for noticing. We added to lines 396–397 that correlation with width and distance from the keel front was negative.*

L360: I suggest to change "seasonal" to "temporal" or "month-long" because you are only considering a limited fraction of the summer season.

**Reply:** *We agree with your comment and substituted "seasonal" to "temporal".*

*We thank the reviewer for the helpful comments and the work they put into the review. Our answers start with "**Reply:**" in bold, following the original reviewers' comments.*

This paper presents the ice ridge melting process obtained from repeated observations using multi-beam sonar carried by ROV, and such observations are extremely rare. The data reveals the relationship between ice ridge melt rate and morphological parameters, the differences from the melt of level ice, and the contributions of ice surface melting, melt-pond discharge, sea ice density, and porosity changes to draft changes. The results are of great significance for the study of heat, matter, and momentum exchange between sea ice and the ocean, as well as algorithms for retrieving sea ice thickness based on satellite altimeters. Therefore, I believe this is a work worth publishing, and recommend considering publishing it after some minor revisions. Here are some special comments:

**Reply:** *We thank you for your positive evaluation of our manuscript. We have done our best to adapt to your main concerns. We have provided detailed replies to all your questions and suggestions below, attempting to provide enough referenced evidence.*

Title: Observations are mainly conducted around the Fram Strait, where the atmospheric and oceanic natures are significantly different from those in the central Arctic Ocean. Therefore, the observation results can only represent that in the observation region. For example, the statistical relationship between the melt rate of ice ridge and the morphological parameters of ice ridge might be significantly different in the central Arctic Ocean. Therefore, it is recommended to focus the topic more on the observational study in "the Fram Strait region".

**Reply:** *We cannot agree that the atmospheric and oceanic conditions of the area of our investigations are significantly different from those of the central Arctic Ocean (and the word "central" is not present in our title). We provide ample evidence for this in the manuscript, as outlined in Section 2.1.*
*First, the sea ice and ridges studied were formed in the central Arctic, and the conditions in the ocean that primarily affect the ice melt are very similar and controlled by the seasonal increase of solar insolation as they would be in the central Arctic. Because the ice ended up in Fram Strait, it does not mean that the conditions changed drastically.*
*Second, the marginal ice zone (MIZ) conditions are met (typically) at sea ice concentrations below 80% (NSIDC; Horvat, 2021, 10.1038/s41467-021-22004-7), while during MOSAiC drift ice concentrations were above 85% (Krumpen et al., 2021).*
*Further, we dedicated a part of our study (lines 101–103) to show that ocean heat fluxes are comparable to those for the Central Arctic (Krishfield, 2005) and for SHEBA (Perovich, 2003). Krishfield (2005) also demonstrated that ocean heat fluxes in the Fram Strait are similar to those in other parts of the Arctic Ocean at comparable latitudes, primarily due to the seasonal increase in solar insolation.*
*In addition, the meteorological conditions during the MOSAiC were defined by Rinke et al. (2021, 10.1525/elementa.2021.00023) as similar to the whole central Arctic, saying that "these summer record warm and moist conditions occurred not only near the Polarstern but over the whole central Arctic". Therefore, we feel using the term "Arctic" is justified in this case given that "Fram Strait" is also located in the Arctic, and the conditions presented here are not only applicable to Fram Strait due to the origin of the ice and since the ocean conditions are similar to what the floe would have experienced in the central basin in summer.*

The statistical relationship between the melt rate of ice ridge and the morphological parameters of ice ridge: More physical explanations are needed, not just showing statistical relationships.

**Reply:** *These are unique observations of ridge melt in themselves; and unfortunately, we do not have a complete set of ocean observations to be able to explain all correlations. Nevertheless, we presented potential factors affecting ridge melt (the same was done by Amundrud et al. and Skyllingstad et al. without further explanation). For the keel draft, we refer to the model and corresponding observation of enhanced turbulence below ridges by Skyllingstad et al. (2003). In brief, ridge keels enhance turbulence and the mixing of seawater with meltwater (lines 202–203). We showed in our previous study how under-ice meltwater can protect thinner ice from ocean heat fluxes by decreasing OHF (Salganik et al., 2022,*

*10.1525/elementa.2022.00035). This is also in agreement with the work on false bottoms from Notz et al. (2003, 10.1029/2001JC001173). We also provide observations of ice bottom temperatures (Fig. 1a), supporting this theory. The second identified parameter, correlated with ridge melt, is the keel width. We mentioned that the effect of the keel width may include two factors. First, wider keels within the same ridge are usually shallower as they are made from approximately the same ice volume. Second, as we also showed in the referenced study (Salganik et al., 2023, 10.1525/elementa.2023.00008), wider keels are usually more consolidated, with most ridge voids located in the vicinity of ridge flanks (lines 251–254). Therefore, wider keels are shallower and more consolidated, which may explain the correlation between keel width and keel melt. Explanations beyond this would be pure speculation and rather points for future research.*

Line 57 "while the ridge keel melted by 1.5 m over two weeks, which translates into an equivalent OHF of 300 W m$^{-2}$, 4.8 times higher than for level ice" --If there is large macroporosity inside the ice ridge, this conversion is not appropriate.

**Reply:** *Here we directly refer to the ocean heat flux value of 298 W m$^{-2}$ estimated by Shestov et al. (2018), as well as to the keel melt reported in their study. Therefore, since the conversion was done by Shestov et al., we only reported the ratio of fluxes in two studies about level ice and ridges. Moreover, a ridge rubble macroporosity of 27 % (ridge R4, June 12) was considered by Shestov et al. (2018) in their estimate of the ocean flux (see Eq. 13), as now mentioned in line 65. (To note also in terms of the next question, these are typical ocean fluxes when actual warm water in the marginal ice zone and vicinity of Atlantic water is encountered.)*

Line 100 "Despite the floe also drifting further south into Fram Strait and getting closer to shallower and warmer Atlantic Water at the same time, the mixed layer and upper ocean conditions still retained their Arctic characteristics"--Note that, it is in the marginal ice zone. The heat content of the oceanic mixed layer has significantly increased, which cannot be considered to be the same as inside the Arctic Ocean, although it is not directly located over the Atlantic warm inflow.

**Reply:** *We disagree. In summer, the heat content of the Arctic summer mixed layer increased due to solar heating, which would be the same primary mechanism as for the summer mixed layer anywhere except in the areas that are exposed to very shallow Atlantic water (see above response); thus, this is still representative of the central basin. This is a typical process for the Central Arctic (Krishfield, 2005). We compared the ocean heat fluxes for MOSAiC, SHEBA (Perovich et al., 2003), and the Central Arctic (Krishfield, 2005) and showed that they are comparable (lines 101–103) and thus representative of the summer mixed layer in the deeper basin as well. This is also noted in the thorough work on oceanography during MOSAiC in Shulz et al. (2023; 10.31223/X5TT2W), which indicate the ocean conditions were similar than in the deeper basin, and ocean heat fluxes stayed low throughout the observations period. We also show that the morphology of the ridge we observed has similar characteristics to other Arctic sea ice ridges (lines 377–381). Given that the ice was formed in the central Arctic, as was the ridge, and the overall lack of such detailed observations in the Arctic, and that other MOSAiC papers, like the paper by Lei et al. cited below, do not make this distinction clearly enough either, we feel it is unwarranted to do so here. We also disagree that our observations were performed in the marginal ice zone. We refer to the estimates of sea-ice concentration of 85–100% from Krumpen et al. (2021), and we show the widely-used 80% threshold of the marginal ice zone in Fig. 1b, which supports that our measurements were located outside of the marginal ice zone, only approaching it.*

1 Level ice melt: The growth and melt rate of level ice are related to ice thickness, and it is recommended to further compare the melt rate with different thicknesses of level ice in the MOSAiC DN region (e.g., Lei et al., 2022). In this way, the results can be more representative and persuasive.
Lei R, Cheng B, Hoppmann M, Zhang F, Zuo G, Hutchings J K, Lin L, Lan M, Wang H, Regnery J, Krumpen T, Haapala J, Rabe B, Perovich D K, Nicolaus M. 2022. Seasonality and timing of sea ice mass balance and heat fluxes in the Arctic transpolar drift during 2019–2020. Elementa: Science of the Anthropocene, 10 (1), doi: https://doi.org/10.1525/elementa.2021.000089.

*Reply: We thank you for this suggestion. Lei et al. (2022) presented bottom melt for 8–9 ice mass balance buoys for 10 July and 20 July and concluded that "lack of statistical significance suggests that the ice melt process was more complex than that for ice growth". At least several of those IMBs were ponded, which makes their estimate of ice surface and total melt less representative (in contrast to the coring measurements we have used that are from the same floe and time). Following your suggestion, we reanalyzed the data from MOSAiC IMBs (see below) and added to lines 277–279 that "Following Lei et al. (2022), we used the data from 10 IMBs installed during MOSAiC to estimate an average bottom melt of 0.17±0.07 m for level ice with an initial thickness of 1.8±0.2 m from 22 June to 20 July, with no significant correlation between the initial ice thickness and bottom melt". This supports our observations of a large spatial variability of ice melt and suggests that the number of point measurements on the order of 10 is not sufficient to study this problem and we added a sentence to this aim in the Conclusions as well (lines 394–396). For sonar measurements, the positive correlation between level ice melt and initial draft in our data is described in lines 198–199.*

[Figure]

Line 183 "the melt of ridge flanks stronger (1.7 times larger regression slope) depended on ice draft in comparison to the keel bottom (Fig. 2b)"-- I cannot get this information from Fig. 2b.

*Reply: We are sorry for this mistake; it should be a reference to Fig. 3b, not 2b. We corrected this error in our manuscript.*

Figure 5c: What does the change in the ice sail surface before the snow completely melts.

*Reply: We thank you for this question and agree that usually the ice surface melt starts after the snow melt is completed. We added a clarification to Fig. 5c caption and line 266 that we present and use the average snow thickness for each of the seven drilling surveys. The sail surface and keel bottom interfaces in Fig. 5c were found using heating-induced temperatures. Unfortunately, the temperature data from the ice mass balance buoy did not allow us to locate the air-snow interface and estimate the corresponding snow thickness accurately; therefore, we opted not to show it. To indicate this difference in types of measurements more clearly in Fig. 5c, we show estimates from ice mass balance buoy (IMB) and drilling using different line types and note this in the caption.*

Line 304 "This difference may be related to the larger (second-year and multiyear) age of SHEBA ridges "-- Why do the age of ice ridges cause such differences?

*Reply: We have now detailed this in the text (lines 341–371). We do not know the exact reasons for such an observation, and given that such observations are still rare, we can only speculate on the exact cause. Second-year ridges are usually fully consolidated, have a smoother shape, and have low salinity (lines 341–343). Each of these factors may contribute to the observed difference in melt rates. We also note the fact that with the very few point measurements taken on SHEBA, it is also possibly a result of undersampling compared to the complete coverage using a multibeam sonar, given the large spatial variability of melt we observed with the sonar surveys (lines 368–371).*